# BMI-adjusted adipose tissue volumes exhibit depot-specific and divergent associations with cardiometabolic diseases

Saaket Agrawal [1,2,3,10], Marcus D. R. Klarqvist [4,10], Nathaniel Diamant [4], Takara L. Stanley[5], Patrick T. Ellinor[1,3], Nehal N. Mehta[6], Anthony Philippakis [4,7], Kenney Ng[8], Melina Claussnitzer[1,2,3], Steven K. Grinspoon[5], Puneet Batra [4] & Amit V. Khera[1,2,3,9] ✉

For any given body mass index (BMI), individuals vary substantially in fat distribution, and this variation may have important implications for cardiometabolic risk. Here, we study disease associations with BMI-independent variation in visceral (VAT), abdominal subcutaneous (ASAT), and gluteofemoral (GFAT) fat depots in 40,032 individuals of the UK Biobank with body MRI. We apply deep learning models based on two-dimensional body MRI projections to enable near-perfect estimation of fat depot volumes ($R^2$ in heldout dataset = 0.978-0.991 for VAT, ASAT, and GFAT). Next, we derive BMI-adjusted metrics for each fat depot (e.g. VAT adjusted for BMI, VATadjBMI) to quantify local adiposity burden. VATadjBMI is associated with increased risk of type 2 diabetes and coronary artery disease, ASATadjBMI is largely neutral, and GFATadjBMI is associated with reduced risk. These results – describing three metabolically distinct fat depots at scale – clarify the cardiometabolic impact of BMI-independent differences in body fat distribution.

Obesity is a leading threat to global public health, with afflicted individuals at increased risk of cardiovascular events, type 2 diabetes, cancer, and severe COVID-19 infection[1–3]. Recent projections suggest that obesity – defined by body mass index (BMI) of at least 30 kg/m² – will affect more than half of the U.S. adult population as early as 2030[4,5].

Although individuals with increased BMI tend to have higher risk of adverse outcomes on average, previous studies have suggested considerable heterogeneity[6–9]. These studies have sought to define markers of "metabolic health" – such as measures of insulin resistance or waist circumference – as drivers of "within BMI-group variation" in cardiometabolic risk[9–11].

Variation in fat distribution is a potential unifying explanation for cardiometabolic risk differences between two individuals with the same BMI[12,13]. Prior studies have suggested that various fat depots have differing metabolic programs, with visceral adipose tissue (VAT) most strongly associated with cardiometabolic risk – but have potential limitations[14–16]. First, most imaging studies to date have been cross-sectional and relatively small – especially those utilizing the gold-standard MRI modality – limiting ability to assess for depot-specific effects across age, sex, and BMI subgroups[12,17–21]. Deep learning models trained on a small set of labeled images and subsequently applied to a larger set of unlabeled images may be one strategy to increase sample

[1]Cardiovascular Disease Initiative, Broad Institute of MIT and Harvard, Cambridge, MA, USA. [2]Center for Genomic Medicine, Department of Medicine, Massachusetts General Hospital, Boston, MA, USA. [3]Department of Medicine, Harvard Medical School, Boston, MA, USA. [4]Data Sciences Platform, Broad Institute of MIT and Harvard, Cambridge, MA, USA. [5]Metabolism Unit, Department of Medicine, Massachusetts General Hospital, Boston, MA, USA. [6]National Heart, Lung, and Blood Institute, National Institutes of Health, Bethesda, MD, USA. [7]Eric and Wendy Schmidt Center, Broad Institute of MIT and Harvard, Cambridge, MA, USA. [8]Center for Computational Health, IBM Research, Cambridge, MA, USA. [9]Verve Therapeutics, Cambridge, MA, USA. [10]These authors contributed equally: Saaket Agrawal, Marcus D. R. Klarqvist. ✉e-mail: avkhera@mgh.harvard.edu

size if models were sufficiently predictive. Second, gluteofemoral adipose tissue (GFAT) – which may serve as an adaptive energy storage depot and a possible modifier of insulin resistance – has not been quantified in most previous imaging studies[18–22]. Third, fat depot volumes tend to be highly correlated with both BMI and one another, making it challenging to isolate depot-specific associations with disease[23].

In this study, we downloaded raw MRI imaging data from 40,032 participants of the UK Biobank and tested the hypothesis that deep learning models can be used to precisely quantify three fat depot volumes: VAT, abdominal subcutaneous adipose tissue (ASAT), and GFAT. We derived measures of local adiposity burden, each fully independent of BMI, and note significant heterogeneity in risk conferred: VAT adjusted for BMI (VATadjBMI) associated with increased risk of type 2 diabetes and coronary artery disease, ASA-TadjBMI largely risk-neutral, and GFATadjBMI associated with protection.

## Results

Among 40,032 participants of the UK Biobank with MRI data available, the median age was 65 years, 51% were female, and 97% were white (Table 1). Median BMI was 26.6 kg/m$^2$ among males and 25.2 kg/m$^2$ among females, and the median waist-hip ratio (WHR) was 0.93 among males, and 0.81 among females. 1,901 individuals had been diagnosed with type 2 diabetes (4.7%) and 1956 with coronary artery disease (4.9%) at the time of imaging assessment. VAT, ASAT, and GFAT volumes were previously quantified in 9040, 9041, and 7754 participants, respectively (Supplementary Data 1 and 2)[20,21,24,25].

**Table 1 | Baseline characteristics of UK Biobank participants at the time of MRI imaging**

|  | Male (N = 19,435) | Female (N = 20,597) |
|---|---|---|
| Age (years) | 66.0 [59.3, 71.3] | 64.1 [58.0, 69.7] |
| **Self-reported ethnicity** | | |
| White | 18,773 (96.6) | 19,936 (96.8) |
| Black | 137 (0.7) | 192 (0.9) |
| East asian | 112 (0.6) | 137 (0.7) |
| South asian | 238 (1.2) | 133 (0.6) |
| Other | 175 (0.9) | 199 (1.0) |
| Systolic blood pressure(mmHg) | 140.5 [130.0, 152.5] | 134.0 [122.0, 147.5] |
| Diastolic blood pressure(mmHg) | 80.5 [74.0, 87.0] | 76.5 [70.0, 83.5] |
| Current smoker | 785 (4.1) | 583 (2.9) |
| Weight (lbs) | 181.3 [164.8, 201.3] | 147.4 [132.2, 166.3] |
| Height (in) | 69.3 [67.8, 71.1] | 64.2 [62.6, 65.8] |
| Body-mass index (kg/m$^2$) | 26.6 [24.4, 29.1] | 25.2 [22.8, 28.5] |
| Waist circumference (cm) | 94.0 [87.0, 101.0] | 81.0 [74.0, 90.0] |
| Hip circumference (cm) | 100.0 [96.0, 105.0] | 100.0 [94.0, 106.0] |
| Waist-to-hip ratio | 0.93 [0.89, 0.98] | 0.81 [0.77, 0.87] |
| **Fat depot volumes** | | |
| Visceral adipose tissue (L) | 4.8 [3.2, 6.4] | 2.3 [1.5, 3.5] |
| Abdominal subcutaneous adipose tissue (L) | 5.4 [4.2, 7.0] | 7.4 [5.6, 9.7] |
| Gluteofemoral adipose tissue (L) | 8.9 [7.5, 10.7] | 10.8 [9.0, 13.1] |
| **Cardiometabolic diseases** | | |
| Type 2 diabetes | 1,264 (6.5%) | 637 (3.1%) |
| Coronary artery disease | 1,542 (7.9%) | 414 (2.0%) |

Continuous variables are reported as medians with interquartile range.

### Machine learning facilitates near-perfect estimation of fat depot volumes

We set out to test whether convolutional neural network models could be adequately predictive of VAT, ASAT, and GFAT volumes to enable prediction at scale. We noted that three-dimensional MRI data for 40,032 individuals represented a substantial data burden with almost 58 million axial slices across all participants, corresponding to >18 terabytes of imaging data – a level of complexity that limits computational feasibility for training deep learning models.

To simplify the imaging input into the convolutional neural networks, we transformed three-dimensional MRI images for each participant into two-dimensional coronal and sagittal projections, hypothesizing that this input would prove adequate for highly accurate fat depot volume prediction despite an 830-fold reduction in data input size (Fig. 1)[26]. Convolutional neural networks – trained on 80% of the participants with fat depots previously quantified – demonstrated near-perfect estimation of each fat depot volume in the 20% of held out individuals (R$^2$ = 0.991, 0.991, and 0.978 for VAT, ASAT, and GFAT, respectively) (Methods, Supplementary Data 3). Similar predictive accuracy was noted across age, sex, BMI, and self-reported ethnicity subgroups, although the sample size was limited in the latter subgroups (Supplementary Data 4). These convolutional neural network models were subsequently applied to the unlabeled remainder of the 40,032 participants to estimate fat depot volumes.

Next, we applied Gradient-weighted Class Activation Mapping (Grad-CAM) to better understand regions of a given MRI projection contributing to predictions of VAT, ASAT, and GFAT volumes[27]. Briefly, Grad-CAM uses gradients entering the final convolutional layer to generate a low-resolution heat map signifying how much a given region contributes to a model's prediction. Separately in males and females, we selected participants from each held out dataset at the 75th, 95th, and 99th percentiles of absolute error and applied Grad-CAM to generate saliency maps. We also selected three participants who were present in all three held out datasets to compare Grad-CAM results for VAT, ASAT, and GFAT. In all cases, Grad-CAM revealed prioritized regions of the MRI projection that were anatomically consistent with the known distribution of VAT, ASAT, and GFAT, even in cases with higher absolute error (Supplementary Figs. 1–4).

### Variation in adipose volumes and association with cardiometabolic diseases

We confirm and extend prior evidence for marked differences in fat depot volume in male versus female participants (Fig. 2a)[28,29]. Median visceral adipose tissue volume was substantially higher in males as compared to females – 4.8 versus 2.3 liters, respectively – while subcutaneous and gluteofemoral depots tended to predominate in females (Table 1). A significant correlation between BMI and all three fat depots was noted – Pearson r ranging from 0.77 to 0.91 – but considerable variation was observed within any clinical BMI category (Fig. 2a, b). Modest variation in the correlation between BMI and each fat depot was noted across self-reported ethnicity groups (Supplementary Data 5).

Adipose tissue volumes were each associated with increased prevalence of type 2 diabetes and coronary artery disease (Supplementary Data 6–7) – as might be expected based on the strength of correlation with BMI – with risk gradient somewhat more pronounced for VAT (Supplementary Data 8). Taking type 2 diabetes as an example, odds ratios per standard deviation increment (OR/SD) were 2.14 (95% CI: 2.05-2.23), 1.69 (95% CI: 1.63-1.75), and 1.48 (95% CI: 1.42-1.54) for VAT, ASAT, and GFAT, respectively.

### BMI-adjusted local fat depots and cardiometabolic disease

To disentangle the unique impact of each fat depot from overall BMI, we next generated measurements of VATadjBMI, ASATadjBMI,

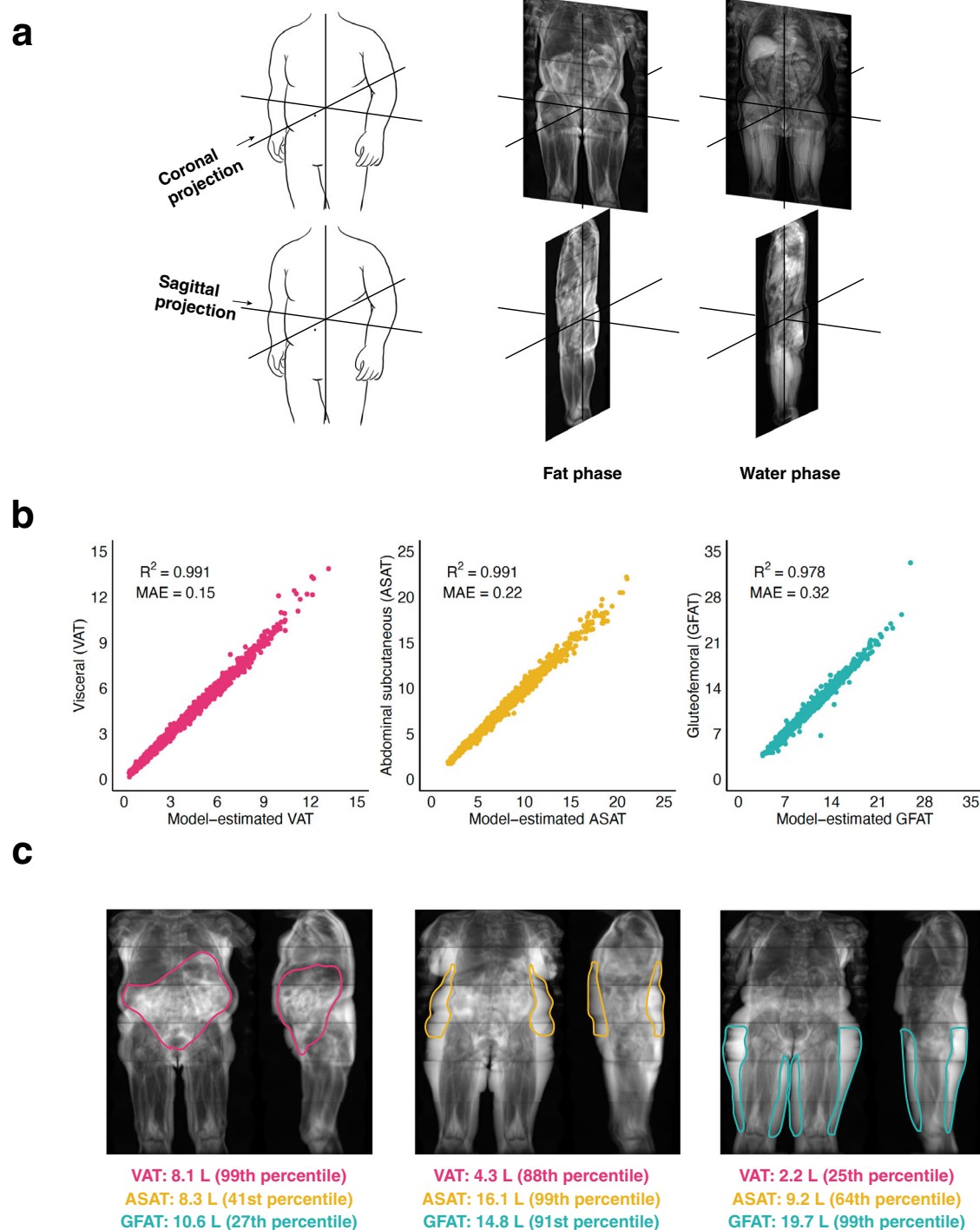

VAT: 8.1 L (99th percentile)
ASAT: 8.3 L (41st percentile)
GFAT: 10.6 L (27th percentile)

VAT: 4.3 L (88th percentile)
ASAT: 16.1 L (99th percentile)
GFAT: 14.8 L (91st percentile)

VAT: 2.2 L (25th percentile)
ASAT: 9.2 L (64th percentile)
GFAT: 19.7 L (99th percentile)

**Fig. 1 | Convolutional neural networks to quantify adipose tissue depots from body MRI images. a** Two-dimensional projections are created by computing the mean pixel intensity along the coronal and sagittal axes. Two images for each participant were used as inputs into the convolutional neural network: one consisting of the coronal and sagittal two-dimensional projections in the fat phase, and another consisting of the same projections in the water phase. **b** Convolutional neural networks trained on two-dimensional MRI projections achieved near-perfect prediction of each fat depot volume in the holdout set (Supplementary Table 3).

**c** Three female participants with similar BMI (ranging from 29.1 to 29.6 kg/m²) but highly discordant fat depot volumes quantified by convolutional neural networks. Fat depot volume percentiles are computed relative to a subgroup of female participants with overweight BMI ($25 \leq BMI < 30$). Note that outlines for each fat depot are drawn as a visual aid for each fat depot and do not reflect segmentation. Abbreviations: VAT, visceral adipose tissue; ASAT, abdominal subcutaneous adipose tissue; GFAT, gluteofemoral adipose tissue.

and GFATadjBMI for each participant by computing sex-specific BMI residuals in 38,680 (97%) of the study population with BMI measurement on the day of MRI imaging available (Supplementary Fig. 5). These residuals reflect the difference in an individual's fat depot volume as compared with that expected based on BMI. These

metrics were fully independent of BMI and largely independent of anthropometric measures and each other (Supplementary Fig. 6). Flexibly modeling BMI with a B-spline basis when computing these residuals yielded similar results (Supplementary Fig. 7, Supplementary Data 9).

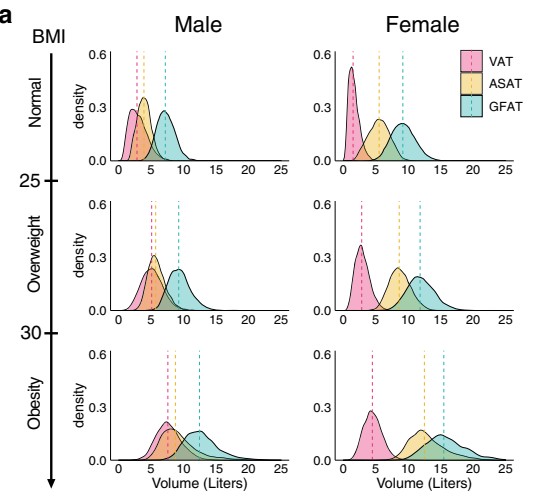
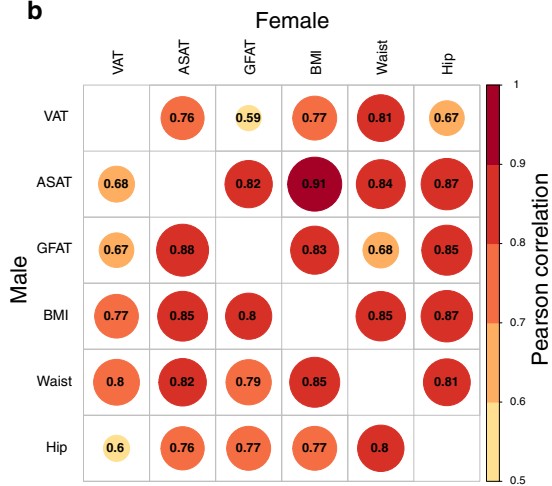

**Fig. 2 | Sex-stratified density plots and correlation plots of visceral, abdominal subcutaneous, and gluteofemoral adipose tissue volumes. a** Sex- and BMI-group specific density plots for visceral adipose tissue (VAT), abdominal subcutaneous adipose tissue (ASAT), and gluteofemoral adipose tissue (GFAT) with dotted lines denoting medians. **b** Sex-stratified correlation plots between VAT, ASAT, GFAT and three anthropometric measures: body mass index (BMI), waist circumference (Waist), and hip circumference (Hip). Analogous plots for BMI-adjusted fat depots are shown in Supplementary Figs. 5–6. Abbreviations: VAT, visceral adipose tissue; ASAT, abdominal subcutaneous adipose tissue; GFAT, gluteofemoral adipose tissue; BMI, body mass index; Waist, waist circumference; Hip, hip circumference.

In contrast to analysis of raw tissue volumes – where each depot was associated with increased risk – significant heterogeneity was noted for BMI-adjusted values. In a mutually adjusted logistic regression model including covariates of age, sex, BMI, and MRI assessment center, we observe that VATadjBMI was associated with increased prevalence of type 2 diabetes – OR/SD 1.49 (95% CI: 1.43–1.55, $P = 9.9 \times 10^{-76}$). By contrast, a largely neutral effect estimate was noted for ASATadjBMI (OR/SD 1.08; 95% CI: 1.03-1.14, $P = 0.002$) and GFATadjBMI volumes were associated with decreased risk (OR/SD 0.75; 95% CI: 0.71-0.79, $P = 6.4 \times 10^{-28}$) (Fig. 3). Effect estimates were largely consistent in subgroups binned by age or sex, with a somewhat more pronounced magnitude of association in participants with BMI less than 25 (Supplementary Figs. 8–9, Supplementary Data 10–12). Within the limits of statistical power owing to small numbers of Black, East Asian, and South Asian participants, we did not detect significant heterogeneity for these associations (p-value for heterogeneity range = 0.83−0.96; Supplementary Data 10–11). A similar pattern was observed for coronary artery disease, where associations for VATadjBMI, ASATadjBMI, and GFATadjBMI were OR/SD 1.17 (95% CI: 1.11–1.22, $P = 3.0 \times 10^{-11}$), 1.00 (95% CI: 0.94–1.05, $P = 0.92$), and 0.89 (95% CI: 0.84–0.94, $P = 3.5 \times 10^{-5}$), respectively. In a sensitivity analysis, we additionally adjusted for weight, height, smoking status, and self-reported ethnicity, finding similar results (Supplementary Data 13). Adjustment for type 2 diabetes status in the coronary artery disease analysis led to comparable results as well.

To better understand the gradients in absolute prevalence rates according to BMI-adjusted fat depots, we calculated standardized estimates for the lowest quintile, quintiles 2–4, and the highest quintile within clinical BMI categories of normal, overweight, and obesity.

Using this approach, we note substantial gradients in the prevalence of cardiometabolic diseases according to local adipose tissue burden, even within clinical BMI categories (Fig. 4, Supplementary Data 14–15). As a representative example, males with normal BMI but VATadjBMI in the highest quintile had a predicted probability of type 2 diabetes of 6.6% (95%CI 5.5–7.9), higher than males with overweight BMI with VATadjBMI in the lowest quintile, in whom probability was 2.7% (95% CI: 2.2–3.4). Among females with obesity, estimates of diabetes ranged from 3.5 to 9.2% across quintiles of VATadjBMI but 7.6 to 3.6% for GFATadjBMI. A similar pattern – with less pronounced gradients – was observed for coronary artery disease.

## BMI-adjusted fat depots and risk of incident cardiometabolic diseases

Over a median follow-up of 2.8 years, 227 (0.6%) and 588 (1.6%) participants with local adiposity metrics available had a new diagnosis of type 2 diabetes or coronary artery disease, respectively. BMI-adjusted fat depots were similarly associated with risk of future disease events in mutually adjusted models. For incident type 2 diabetes, hazard ratios per SD increase (HR/SD) were 1.45 (95% CI: 1.30–1.61, $P = 1.3 \times 10^{-11}$), 0.96 (95% CI: 0.84–1.08, $P = 0.49$), and 0.84 (95% CI: 0.74–0.95, $P = 0.005$) for VATadjBMI, ASATadjBMI, and GFATadjBMI respectively (Table 2). For incident coronary artery disease, HR/SD were 1.17 (95% CI: 1.08–1.26, $P = 8.1 \times 10^{-5}$), 1.04 (95% CI: 0.95–1.14, $P = 0.41$), and 0.91 (95% CI: 0.83–1.00, $P = 0.05$) for VATadjBMI, ASATadjBMI, and GFATadjBMI respectively.

## Association of lifestyle habits with fat depots

Of the 40,032 studied participants, 39,530 had self-reported data regarding diet and physical activity available at the time of imaging (Supplementary Data 16). Participants were categorized as following either an ideal or poor diet and either ideal, intermediate, or poor physical activity on the basis of previously defined criteria[30]. We studied associations between diet and physical activity categories with each BMI-adjusted fat depot in linear regressions adjusted for age, sex, smoking status, and MRI assessment center. Ideal diet was associated with reduced VATadjBMI (beta = −0.15 SDs; 95% CI −0.18 - −0.13, $P = 6.8 \times 10^{-39}$), with weaker associations noted with ASATadjBMI (beta = −0.04 SDs; 95% CI: −0.06 - −0.02, P = 0.001) and GFATadjBMI (beta = −0.03 SDs; 95% CI: −0.05–0.00, $P = 0.03$) (Supplementary Data 17). Intermediate versus poor physical activity revealed a more symmetric pattern with reduced VATadjBMI (beta = −0.13 SDs; 95% CI: −0.17−(−0.09), $P = 5.3 \times 10^{-11}$), ASATadjBMI (beta = −0.07 SDs; 95% CI: −0.11−(−0.03), and GFATadjBMI (beta = −0.08 SDs; 95% CI: −0.12−(−0.04)). Ideal versus poor physical activity showed a similar pattern with an amplified effect. Similar patterns were observed in models examining associations with BMI-unadjusted VAT, ASAT, and GFAT.

## Discussion

In this study, we demonstrated that a deep learning approach based on two-dimensional MRI projections is adequately predictive to quantify

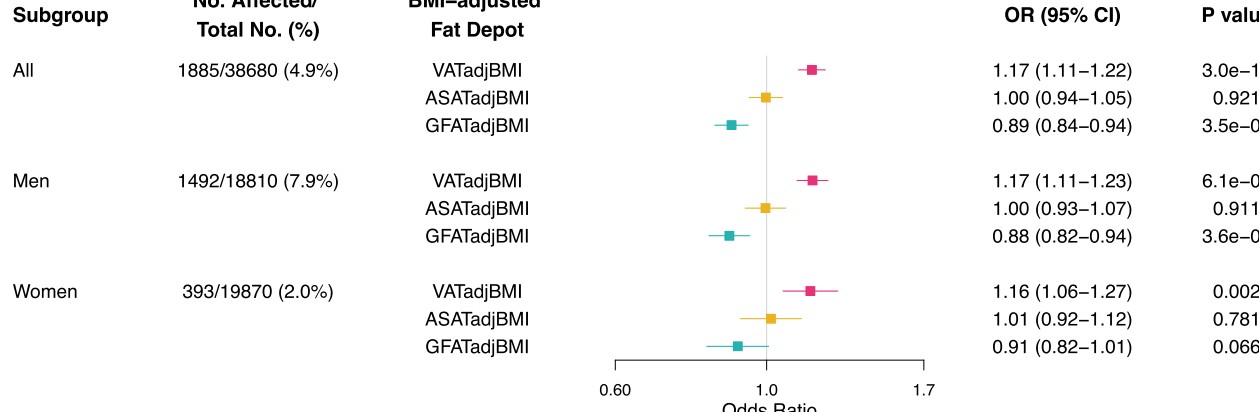

**Fig. 3 | Association of body-mass index adjusted fat depots with type 2 diabetes and coronary artery disease.** Odds ratios per standard deviation with 95% confidence intervals are shown for prevalent type 2 diabetes and coronary artery disease. Sample sizes for each model are shown as the denominator in the second column. P-values correspond to two-sided tests for the indicated independent variable in an adjusted logistic regression. Logistic regression models were adjusted for age, sex (except in sex subgroup analyses), BMI, the other two fat depots (e.g. ASATadjBMI and GFATadjBMI for VATadjBMI), and MRI imaging center. Source data are provided as a Source Data file. Abbreviations: VATadjBMI, visceral adipose tissue adjusted for body mass index (BMI); ASATadjBMI, abdominal subcutaneous adipose tissue adjusted for BMI; GFATadjBMI, gluteofemoral adipose tissue adjusted for BMI.

VAT, ASAT, and GFAT volumes at scale. By then moving away from raw fat depot volumes – which are driven largely by BMI and overall adiposity – to BMI-adjusted measurements, we demonstrated a consistent trend of VATadjBMI associated with increased risk of type 2 diabetes and coronary artery disease, ASATadjBMI largely risk-neutral, and GFATadjBMI conferring protection. These results have at least four implications.

First, machine learning can enable insights from large-scale data repositories of difficult-to-measure phenotypes. In this study, convolutional neural network models were used to precisely measure fat depot measurements from MRI images, considered the gold standard modality for the volumetric measurement of adipose tissue[16,31]. Hypothesis-informed simplification of the input data – in this study moving from three-dimensional MRI images to two-dimensional MRI projections – was necessary to ensure computational feasibility. This work adds to several recent studies of machine learning-derived phenotypes, including aortic size, liver fat, and cardiac trabecular structure[32–34]. Although population-based assessment of fat distribution using MRI is unlikely to be practical, these results lay the scientific foundation for efforts to quantify such measures using other data – such as DEXA images or abdominal CT scans already embedded in the electronic medical record for some patients – or even static images of body silhouette, as might conceivably be obtained with a smartphone camera[35,36]. Abdominal

imaging may also be useful for learning hidden variables of biological significance, such as age[37].

Second, these results support a growing appreciation that various fat depots – rather than serving as an agnostic sink for energy storage – have distinct metabolic profiles. Previous work has noted significant functional differences in adipocytes according to specific fat depot, ascribed in part to site-specific expression of developmental genes associated with adipogenesis[38,39]. While VAT tends to be the primary site for immediate storage of dietary-derived fat via adipocyte hypertrophy and has a higher rate of lipid turnover, GFAT is a more stable fat depot that primarily expands via adipocyte hyperplasia and may spare expansion of harmful visceral or ectopic fat depots. These and other studies support a natural order of fat deposition, whereby a primary driver of high VAT in specific individuals may reflect an inability to adequately expand ASAT or GFAT depots[13,40]. In rare Mendelian lipodystrophies – as occurs in individuals who harbor pathogenic *LMNA* mutations – an extreme example of this paradigm leads to marked reduction of ASAT and GFAT but increased VAT and increased rates of severe insulin resistance[41]. Whether individuals in the extreme tails of low GFATadjBMI and ASATadjBMI or high VATadjBMI might be enriched for genetic perturbations in lipodystrophy genes or the inherited component to these metrics is largely 'polygenic' – due to the aggregate effects of many common DNA variants, each of modest effect size – warrants further study[42–44]. Sex differences will also be important to

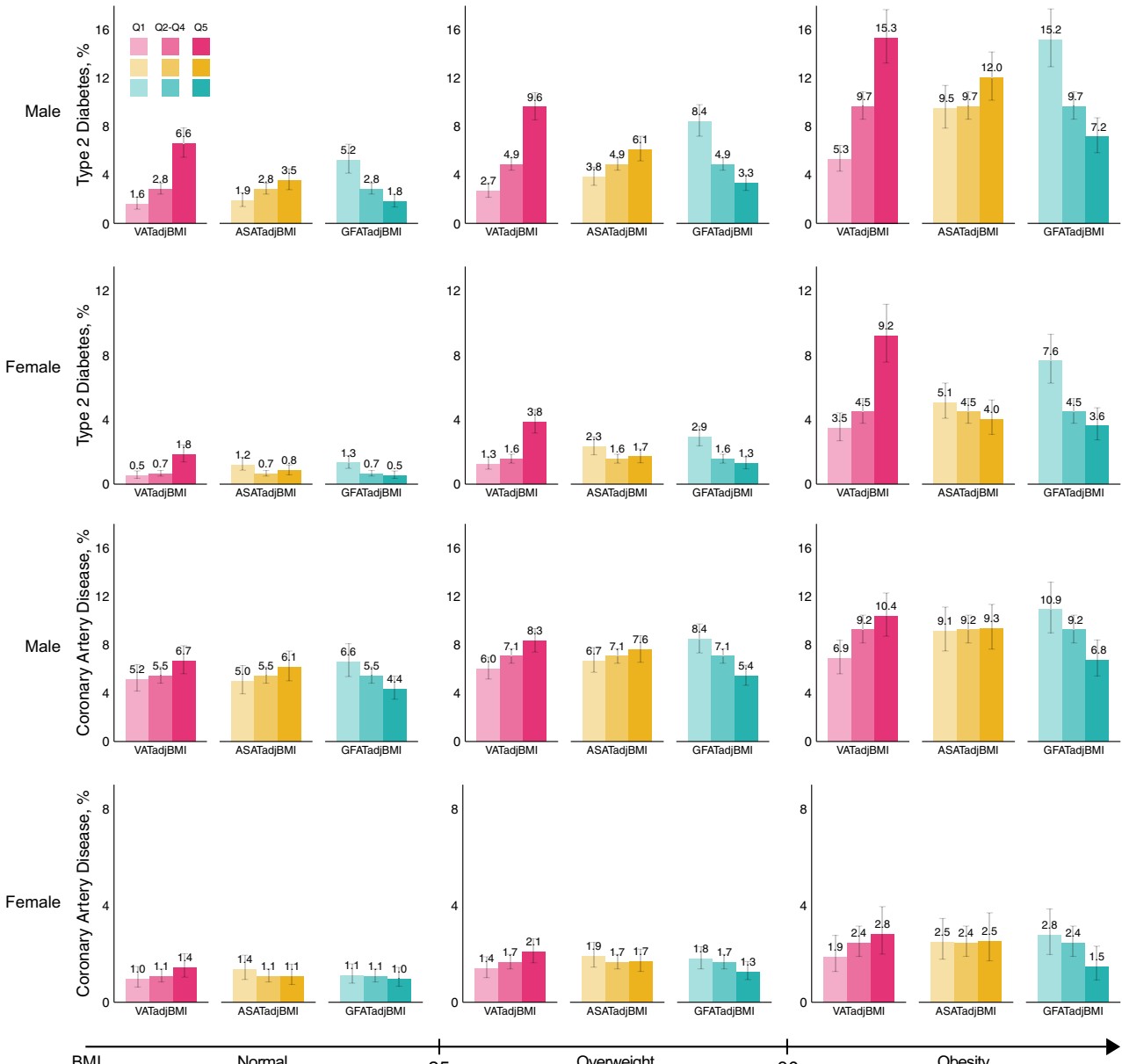

**Fig. 4 | Standardized prevalence of type 2 diabetes and coronary artery disease, according to quintiles of body-mass index adjusted fat depot and body-mass index strata.** Standardized prevalence with 95% confidence intervals are reported from sex-stratified logistic regressions including age, BMI, MRI imaging center, sex-specific quintiles of three local adiposity metrics (VATadjBMI, ASATadjBMI, GFATadjBMI), and interaction terms between BMI and each of the local adiposity metrics. 18,810 male participants and 19,870 female participants were used for each logistic regression model, respectively. For each fat depot, the three bars from lightest to darkest represent the bottom quintile, quintiles 2–4, and the top quintile of the BMI-adjusted fat depot in question, respectively. Median body-mass index was 25.9 kg/m² with 15,446 (39.9%) individuals with BMI < 25, 16,179 (41.8%) with 25 ≤ BMI < 30, and 7055 (18.2%) with BMI ≥ 30. Source data are provided as a Source Data file. Abbreviations: BMI, body mass index; VATadjBMI, visceral adipose tissue adjusted for BMI; ASATadjBMI, abdominal subcutaneous adipose tissue adjusted for BMI; GFATadjBMI, gluteofemoral adipose tissue adjusted for BMI.

consider in future studies on local adiposity – for example, here we demonstrate that ASATadjBMI and GFATadjBMI are more correlated in male participants than in female participants, which may point to sex-dependent fat depot specificity.

Third, changes in measures of local adiposity – independent of weight and body-mass index – may serve as reliable proxies of cardiometabolic benefits of a given intervention, and warrant consideration as additional endpoints for future clinical trials. Most studies to date of obesity interventions have focused on reduction in overall weight or BMI as the primary outcome, consistent with FDA regulatory guidance[45]. However, at least two classes of drugs appear to have a selective VAT reduction effect in clinical trials: thiazolidinediones and

a synthetic form of growth hormone-releasing hormone[46,47]. Whether these therapies might be repurposed from their original indications – type 2 diabetes and HIV-associated lipodystrophy – or new agents might prove useful in a subset of individuals with VAT-driven increases in cardiometabolic risk warrants further study. In such studies, "adjBMI" or similar measures of local adiposity may prove useful for quantifying BMI-independent changes in fat distribution. Considering measures of local adiposity may be particularly important for individuals with normal or low BMI – in this study, we observed a trend of amplified associations with type 2 diabetes in participants with BMI less than 25 kg/m², consistent with a prior study examining the association of waist circumference and waist-hip ratio with mortality[48].

**Table 2 | Association of BMI-adjusted fat depot volumes with incident disease**

| Disease | No. events / Total no. at risk (%) | BMI-adjusted fat depot | HR (95% CI) | *P*-value |
|---|---|---|---|---|
| Type 2 Diabetes | 227/36,837 (0.6) | VATadjBMI | 1.45 (1.30–1.61) | $1.3 \times 10^{-11}$ |
| | | ASATadjBMI | 0.96 (0.84–1.08) | 0.49 |
| | | GFATadjBMI | 0.84 (0.74–0.95) | 0.005 |
| Coronary artery disease | 588/36,786 (1.6) | VATadjBMI | 1.17 (1.08–1.26) | $8.1 \times 10^{-5}$ |
| | | ASATadjBMI | 1.04 (0.95–1.14) | 0.41 |
| | | GFATadjBMI | 0.91 (0.83–1.00) | 0.05 |

Hazard ratios with 95% CI in parentheses are shown for VATadjBMI, ASATadjBMI, and GFATadjBMI in Cox proportional-hazard models adjusted for age, sex, BMI, the other two fat depots, and MRI imaging center. P-values correspond to two-sided tests for the indicated independent variable in the adjusted models. Median follow-up time for both incident type 2 diabetes and coronary artery disease was 2.8 years from the date of imaging. Note that two participants in the prevalent disease analyses are not included in incident disease analyses because they withdrew consent in the interim period.

Fourth, although our data suggests similar performance of our deep learning models across self-reported ethnicity subgroups, we were underpowered to study disease associations in non-White subgroups. Additional validation across ancestrally and geographically diverse populations would be of considerable value, especially given prior evidence of significant variability in fat distribution indices across ethnicity groups[49,50]. An important example relates to the South Asian population, where abnormal fat distribution has been postulated as a key driver of the markedly increased rates of cardiovascular disease and diabetes observed, often in the context of a relatively normal BMI[51,52].

Our study has several limitations. First, this study was a cross-sectional analysis of individuals with a median age of 65 years at time of imaging. Future studies of individuals across the lifespan – especially those that include repeat imaging assessments – are warranted. Second, although we note striking associations of BMI-adjusted fat depots with cardiometabolic disease, these observational data do not definitely prove causation or that modification of fat distribution will lead to therapeutic gain. Third, while two-dimensional MRI projections are a useful simplification of three-dimensional MRI images for the task of predicting adipose tissue compartment volumes, they are unlikely to be appropriate for predicting "density-like quantities" such as liver fat percentage, where a single axial cross-section performs well[33]. This highlights the importance of choosing an appropriate simplification for the desired task. Fourth, while we were able to achieve good performance using CNN-based regression models and saliency mapping results were anatomically reasonable, we were unable to directly compare our approach to segmentation-based models.

In conclusion, we used a machine learning approach based on two-dimensional projections of body MRI data to compute VAT, ASAT, and GFAT volumes at scale in 40,032 individuals of the UK Biobank. BMI-adjusted fat depot measurements displayed divergent associations with cardiometabolic diseases and were shown to alter risk within BMI subgroups. These BMI-adjusted metrics may serve as useful additional endpoints for obesity interventions to more completely capture metabolic health associated with body composition.

## Methods
### Study population
The UK Biobank is an observational study that enrolled over 500,000 individuals between the ages of 40 and 69 years between 2006 and 2010, of whom 43,531 underwent body MRI imaging between 2014 and 2020 as part of an imaging substudy[53,54]. Images were acquired using the Dixon method, an MRI sequence that can be used to isolate fat signals from water signal[55]. Each participant's MRI data consisted of 244 axial slices acquired from the neck to the knees in four sequences: in-phase, out-of-phase, fat-only, and water-only. After the exclusion of 3489 (8.0%) imaging scans based on technical problems or artifacts, 40,032 participants remained for analysis, 19,435 males and 20,597 females (Supplementary Methods). This analysis of data from the UK Biobank was approved by the

Mass General Brigham institutional review board and was performed under UK Biobank application #7089.

### Machine learning to measure fat depot volumes
Among the 40,032 individuals with MRI imaging data available, a subset had visceral adipose tissue (VAT) volume, abdominal subcutaneous adipose tissue (ASAT) volume, and total adipose tissue (TAT) volume between the top of vertebrae T9 and the bottom of the thigh muscles, quantified and made available as previously described ($N = 9040$, 9041, 7754 participants, respectively)[20,21,24,25]. Gluteofemoral adipose tissue (GFAT) volume was derived by computing the difference between TAT and the sum of VAT and ASAT (Supplementary Methods). For each participant, we transformed three-dimensional MRI images into two-dimensional coronal and sagittal projections by computing the mean intensity projection in each orientation. For example, a given pixel on a coronal two-dimensional projection represents the mean intensity across all pixels making up a line oriented in the anterior-posterior direction perpendicular to the coronal plane (Supplementary Methods). This procedure was done for the fat-only and water-only MRI sequences, and the resulting images were jointly used as the imaging input for a given participant.

Individuals with previously quantified fat depot volumes were randomly split into 80% for training and a 20% holdout sample for testing. For each of VAT, ASAT, and GFAT, a CNN was trained on a pair of fat phase and water phase MRI images to predict each fat depot volume, where each image was composed of (a) a coronal two-dimensional projection and (b) a sagittal two-dimensional projection of the body MRI. Each CNN was developed with the DenseNet-121 architecture pre-trained on ImageNet as the base model[56,57]. The last dense block output was flattened using a global average pooling layer and then fed into three fully connected layers of size 64, 256, and 1, with the last layer having no activation function (linear mapping). All other activation functions use the ReLU non-linearity. All models were trained using the Adam optimizer with a learning rate set to a cosine decay policy decaying from 0.001 to 0 over 100 epochs, a shrinkage loss function using the hyperparameters $a = 10.0$ and $c = 0.2$, and a batch size of 32[58,59]. For all training data, the following augmentations (random permutations of the training images) were applied: random shifts in the *XY*-plane by up to ±16 pixels, rotations by up to ±5 degrees around its center axis, and the coronal view horizontally flipped with a probability of 50%. Each view (coronal and sagittal) were separately pre-normalized by its *z*-score (0 mean, standard deviation of 1), followed by joint normalization following concatenation side-by-side.

A five-fold cross-validation scheme was used within the 80% training data set. Performance was determined in a 20% holdout sample that was unseen to the model prior to evaluation. The five folds were used to determine the mean and standard deviation of performance metrics, then a single fold was randomly selected to take forward for predicting fat depot volumes in the remaining participants with raw MRI imaging data but without labels. Additional information can be found in the Supplementary Methods.

## Saliency maps

We used Gradient-weighted Class Activation Mapping (Grad-CAM)[27] to generate saliency maps for selected participants to obtain "visual explanations" for decisions from our CNN-based regression models used for estimating fat depot volumes. Briefly, Grad-CAM uses the gradients flowing into the final convolutional layer to produce a low-resolution localization heat map highlighting important regions (red) and less important regions (blue). In other words, the importance signifies how much a specific area contributes to the overall prediction.

## Cardiometabolic disease definitions

Type 2 diabetes was defined on the basis of ICD-10 codes, self-report during a verbal interview with a trained nurse, use of diabetes medication, or a glycated hemoglobin greater than or equal to 6.5% before the date of imaging. Coronary artery disease was defined as myocardial infarction, angina, coronary revascularization, or death from coronary causes as determined on the basis of ICD-10 codes, ICD-9 codes, OPCS-4 surgical codes, nurse interview, and national death registries.

## Statistical analysis

We generated BMI-adjusted fat depot measurements by computing residuals from sex-specific linear regression models using BMI to predict each fat depot volume, analogous to prior studies of waist-hip ratio adjusted for BMI[60,61]. Logistic regression models were used to test the association of BMI-adjusted fat depot measurements with prevalent disease in models adjusted for age, sex (except in sex subgroup analyses), BMI, the other two fat depots (e.g. ASATadjBMI and GFATadjBMI for VATadjBMI), and MRI imaging center. Cox proportional-hazard models with the same covariates were used to test associations of BMI-adjusted fat depots with incident type 2 diabetes and coronary artery disease. To predict the gradient of prevalent disease across clinical categories, we used logistic regression models separately in males and females including age, BMI, sex-specific quintiles of VATadjBMI, ASATadjBMI, GFATadjBMI, MRI imaging center, and interaction terms between the local adiposity quintiles and BMI. Models were standardized to the median of all predictor variables (except for the MRI imaging center variable, where the mean was used). Effect sizes are reported per sex-specific standard deviation.

All analyses were performed with the use of R software, version 3.6.0 (R Project for Statistical Computing).

## Reporting summary

Further information on research design is available in the Nature Portfolio Reporting Summary linked to this article.

## Data availability

This research has been conducted using the UK Biobank Resource under Application Number #7089. The raw UK Biobank data - including the anthropometric data reported here - are made available to researchers from universities and other research institutions with research inquiries following IRB and UK Biobank approval (https://www.ukbiobank.ac.uk/enable-your-research/apply-for-access). Visceral, abdominal subcutaneous, and gluteofemoral adipose tissue volume predictions have been submitted to the UK Biobank and will be available for download by researchers (https://www.ukbiobank.ac.uk/enable-your-research/research-analysis-platform). All other data generated in the study are available in the Supplementary Data. Source data for Figs. 3 and 4, Supplementary Fig. 8 and Supplementary Fig. 9 are provided as a Source Data files. Source data are provided with this paper.

## Code availability

Representative code for this work is made available at the following Github repository: https://github.com/broadinstitute/ml4h/tree/master/model_zoo/adiposity_mlandepi.

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

## Acknowledgements

The authors thank Mary O'Reilly of the Broad Institute's Pattern data visualization team for assistance in graphic and visual design. This work was supported by the Sarnoff Cardiovascular Research Foundation Fellowship (to S.A.), grants 1K08HG010155 and 1U01HG011719 (to A.V.K.) from the National Human Genome Research Institute, a Hassenfeld Scholar Award from Massachusetts General Hospital (to A.V.K.), a Merkin Institute Fellowship from the Broad Institute of MIT and Harvard (to A.V.K.), a sponsored research agreement from IBM Research to the Broad Institute of MIT and Harvard (P.T.E., A.P., P.B., A.V.K.), grant R01DK063639 to S.K.G. and by grants M01-RR-01066 and 1 UL1 RR025758-01 to the Harvard Clinical and Translational Science Center from the National Center for Research Resources and the Nutrition Obesity Research Center, Harvard University (National Institutes of

Health grant P30 DK40561). M.D.R.K., S.A., P.B., and A.V.K. are listed as co-inventors on a patent application for the use of imaging data in assessing body fat distribution and associated cardiometabolic risk.

## Author contributions

S.A., M.D.R.K., P.B., and A.V.K. conceived and designed the study. S.A., M.D.R.K., N.D., and T.L.S. acquired, analyzed, and interpreted the data. S.A. and M.D.R.K. drafted the manuscript. S.A., M.D.R.K., N.D., T.L.S., P.T.E., N.N.M., A.P., K.N., M.C., S.K.G., P.B., and A.V.K. critically revised the manuscript for important intellectual content.

## Competing interests

S.A. has served as scientific consultant for Third Rock Ventures. M.D.R.K., N.D., A.P., and P.B. are supported by grants from Bayer AG applying machine learning in cardiovascular disease. T.L.S. has served on an advisory board for Theratechnologies and RosVivo Therapeutics and has received grant funding to her institution from Pfizer, Inc. P.T.E. receives sponsored research support from Bayer AG and IBM and has consulted for Bayer AG, Novartis, MyoKardia and Quest Diagnostics. A.P. is also employed as a Venture Partner at GV and consulted for Novartis; and has received funding from Intel, Verily and MSFT. K.N. is an employee of IBM Research. P.B. serves as a consultant for Novartis. S.K.G has consulted for and received research funds from Theratechnologies and Viiv through his institution. He has received research funds from KOWA and Gilead unrelated to this project also through his institution. A.V.K. is an employee and holds equity in Verve Therapeutics; has served as a scientific advisor to Amgen, Maze Therapeutics, Navitor Pharmaceuticals, Sarepta Therapeutics, Novartis, Silence Therapeutics, Korro Bio, Veritas International, Color Health, Third Rock Ventures, Illumina, Foresite Labs, and Columbia University (NIH); received speaking fees from Illumina, MedGenome, Amgen, and the Novartis Institute for Biomedical Research; and received a sponsored research agreement from IBM Research. The remaining authors declare no competing interests.
