## [Peer Review File · Nature Communications]

Reviewers' Comments:

Reviewer #1:

Remarks to the Author:

The authors used a machine learning approach based on two-dimensional projections of body MRI data to compute VAT, ASAT, and GFAT volumes at scale in 40,032 individuals of the UK Biobank. They found that BMI-adjusted fat depot measurements displayed divergent associations with cardiometabolic diseases and were shown to alter risk within BMI subgroups. The authors concluded that these BMI-adjusted metrics may serve as useful additional endpoints for obesity interventions to more completely capture metabolic health associated with body composition. This is a very well performed study that used an innovative deep learning approach to better handle large image-based body fat composition datasets and that provides interesting information about relationships of important fat depots with type 2 diabetes and coronary artery disease.

Comments:

1. While VAT and SAT volumes were previously quantified in similar numbers of people (9040 and 9041), GFAT (depending on availability of TAT measurement) volume was only quantified in 7754 people. What was the main reason for this difference? The authors should also report in the supplementary information the anthropometrics and cardiometabolic parameters of the people who had VAT and SAT, but not GFAT (TAT), measurements.
2. Table 1: for variables that are not normally distributed the authors should provide the medians and the interquartile range of the data.
3. Because about 96.7% of the subjects studied were white, and body fat distribution and the relationships of body fat distribution with cardiometabolic diseases largely differ among different ethnic groups, the authors need to address the limitation of the study that relationships identified in the UK Biobank may not easily apply to other ethnic groups.
4. Sex- and BMI-group specific density plots for VAT, ASAT and GFAT are shown in the figure 2A and prevalence of type 2 diabetes and coronary artery disease, according to quintiles of body-mass index, are shown in the figure 4. Because the East Asians and South Asians studied have different BMI cut-off values for overweight and obesity, than the presented cut-off values, the authors should take this into account in their analyses.
5. ASATadjBMI and GFATadjBMI correlated quite strongly with each other in men, and to a much lesser extent in women. The authors should address this point in the discussion and hypothesize about possible mechanisms explaining this difference.
6. The authors should also discuss why higher VATadjBMI and lower GFATadjBMI most strongly associated with type 2 diabetes and coronary artery disease, in people with a BMI <25, compared to people with overweight or obesity. These relationships may predominantly be driven by a lipodystrophy-like phenotype in people with very low total fat mass.
7. In the line 158, shouldn't it read "... in whom probability was 5.3% (95%CI 4.3-6.4)."?
8. Typo in line 125.
9. Line 211: the paper referenced as number 45 does not support the statement that "... consistent with FDA regulatory guidance." Also, the papers referenced as 46 and 47 don't support the statement about pharmacological treatment of VAT.

Reviewer #2:

Remarks to the Author:

Association of machine learning-derived measures of body fat distribution with cardiometabolic diseases in >40,000 individuals

We applaud the authors for developing machine learning approaches to uncover digital readouts of body fat and correlating them with cardio metabolic diseases. Many appreciate the relationship between different measures of adiposity capture different types of metabolic risk, and this approach is one way to dissect this biology. The manuscript can be much improved:

- 1.) Despite the strong correlation with different adiposity measures (Figure 1/2), is the algorithm detecting new biology? UK Biobank contains genetic data, as the authors know well, that could be correlated against their new predictor; the genetic architecture can also be compared between the

new predictors and the existing fat measures.

2.) Related to 1, measures such as diet and physical activity are also available - are these modifiable factors correlated with the new predictors? Are these associations heterogeneous?

3.) Little information is provided about the deep learning approach; the parameters used and the methods for cross validation; please include. While "black boxes", methods such as activation maps can describe what the algorithm is focusing on. This reviewer encourages the use of existing libraries to better understand the predictions and/or the non-concordant data.

4.) The authors leave a lot of room to describe clinical utility of these new readouts: what is meant by "additional endpoints for obesity interventions..." Can the authors describe a study to examine how interventions might reflect changes in these image readouts that would be different than that of measuring the phenotypes they trained on?

5.) The authors mention limited age windows of surveillance; however, others have shown the ability for modalities such as MRIs to predict age in the UK Biobank (Le Goallec 2021, Nature Comm; Jonsson ,2019) and may be residually confounded. Have the authors considered a predictor that stratifies by age to get at question posed in 1.)?

Reviewer #3:

Remarks to the Author:

In their manuscript entitled "Association of machine learning-derived measures of body fat distribution with cardiometabolic diseases in >40,000 individuals" the authors describe the results of the analysis of adipose tissue distribution from MRI data in participants from the UK Biobank study.

Overall, the topic of this work - the analysis of large-scale epidemiological imaging data - is highly relevant and important for the scientific community as well as potentially for medical practice.

In this work, the authors provide mainly provide the following two core contributions:

- (i) simplification and optimization of automated quantification of adipose tissue compartments
- (ii) correlation of adipose tissue distribution with a limited number of clinical endpoints, specifically the presence of type 2 diabetes.

Overall the manuscript is clearly structured and well-written.

However, there are major and minor limitations that need to be addressed:

Major points:

I would like to encourage the authors to perform more detailed and thorough analyses. The large-scale data the authors were able to access offers a unique opportunity for scientific analyses but also demands a particular level of analytic effort. This includes:

1) The statistical analyses presented in this work remain partly superficial and inconsistent. First, the authors treat different factors of variance differently without clear reason. Thus, BMI is directly adjusted for in adjusted adipose tissue volumes, but potentially equally important factors such as age are only considered as subgroups. It could e.g. very well be that the computed VATadjBMI just reflects changes of VAT with age and is therefore only indirectly associated with type 2 diabetes (with age as a confounder). Other factors, such as racial background, body height and weight (BMI as an aggregate measure loses some information), may similarly act as confounding factors. Only after correction for these and other potential factors of variation is it possible to hypothesize an independent association between adipose tissue distribution and cardiovascular disease.

2) While the simplification of adipose tissue quantification made by the authors (from 3D to 2D

projections) may allow for faster analyses, it has the major disadvantage that crucial information that is present in the image volumes is discarded. In the context of this work, the used imaging data in principle allow for quantification of further adipose tissue compartments including, e.g., pericardial adipose tissue, kidney hilum adipose tissue as well as liver and pancreas fat content. The analysis of data from large-scale population studies such as the UK Biobank should utilize all relevant information that is available in order to provide a complete picture of the data. The additional effort is justified considering the potential impact of these results. Thus, although technically interesting, the proposed simplified analysis can be seen as an oversimplification.

Further points:

3) Title: the data source (UK Biobank) should be mentioned in the title

4) Methods: How was quality assurance performed on the test set? The proposed adipose tissue quantification method - in contrast to segmentation-based methods - has the drawback of limited interpretability. How were image volumes with wrong automated adipose tissue estimates identified?

5) Results: Did the authors have access to more detailed temporal information regarding the onset of cardiovascular disease. It would be of high interest to look into the correlation between adipose tissue volumes and disease progression if possible.

6) Figures: Figure 1C is somehow misleading as it suggest segmentation of adipose tissue compartments, which is however not done by the proposed algorithm. It should be clarified that the provided regions were hand-drawn for illustration purposes.

7) Methods: It could be interesting to look into gradient-based saliency maps of the adipose tissue segmentation models in order to assess whether estimates are based on the respective anatomic regions or maybe based on completely different information.

8) Methods: What justifies linear adjustment of adipose tissue volumes for BMI? Was a linear relationship observed?

9) Methods: The observed clinical endpoints, specifically type 2 diabetes may themselves be a confounder for associations with other clinical endpoints (e.g. the association of VAT depots with cardiac disease may be mediated by type two diabetes). These potential effects should be considered in the statistical analysis.

Response to Referees' Comments for *NCOMMS-22-30756*
“Association of machine learning-derived measures of body fat distribution with cardiometabolic diseases in >40,000 individuals”

Response to Reviewer #1:

The authors used a machine learning approach based on two-dimensional projections of body MRI data to compute VAT, ASAT, and GFAT volumes at scale in 40,032 individuals of the UK Biobank. They found that BMI-adjusted fat depot measurements displayed divergent associations with cardiometabolic diseases and were shown to alter risk within BMI subgroups. The authors concluded that these BMI-adjusted metrics may serve as useful additional endpoints for obesity interventions to more completely capture metabolic health associated with body composition.

This is a very well performed study that used an innovative deep learning approach to better handle large image-based body fat composition datasets and that provides interesting information about relationships of important fat depots with type 2 diabetes and coronary artery disease.

Author Response: We appreciate and agree with the summary above.

Comments:

1. While VAT and SAT volumes were previously quantified in similar numbers of people (9040 and 9041), GFAT (depending on availability of TAT measurement) volume was only quantified in 7754 people. What was the main reason for this difference? The authors should also report in the supplementary information the anthropometrics and cardiometabolic parameters of the people who had VAT and SAT, but not GFAT (TAT), measurements.

Author Response: We agree that this issue warrants further clarification. The visceral adipose tissue volume (UK Biobank Field ID: 22407), abdominal subcutaneous adipose tissue volume (field 22408), and total adipose tissue volume (between the bottom of the thigh muscles to the top of vertebrae T9, field 22415) values used as truth labels in this study were available for download from the UK Biobank research portal. Their derivation from body MRI is outlined in two publications.^{1,2} At the time of downloading these data for this work, there were fewer participants with data for total adipose tissue volume; this may have been because publications that were using these data were prioritizing visceral adipose tissue volume and abdominal subcutaneous adipose tissue volume for study, although no metadata from the UK Biobank were available that explained this difference.^{3,4}

We have included a new Supplementary Data table showing baseline characteristics stratified by membership in this group, copied below. Members of this 1,286 participant subgroup were less likely to be female, explaining some differences in anthropometry and prevalent diseases. Given that our models perform nearly identically with good accuracy ($R^2 > 96.5\%$) across sex subgroups, we believe this is unlikely to have meaningfully impacted results.

Supplementary Data 1 Baseline characteristics stratified by truth label status

Continuous variables are reported as medians with interquartile ranges.

Note that ASAT and VAT truth label sets perfectly overlap, except for 1 participant who had ASAT, but not VAT.

	No VAT or GFAT Truth Labels	Truth Label Available for VAT and GFAT	Truth Label Available for VAT, but not GFAT
n	30992	7754	1286
Age (years)	65.5 [59.1, 71.0]	63.4 [56.8, 68.3]	64.0 [57.0, 69.0]
Female (%)	15950 (51.5)	4161 (53.7)	486 (37.8)
Self-reported ethnicity			
White	29970 (96.7)	7498 (96.7)	1241 (96.5)
Black	257 (0.8)	58 (0.7)	14 (1.1)
East Asian	191 (0.6)	53 (0.7)	5 (0.4)
South Asian	284 (0.9)	76 (1.0)	11 (0.9)
Other	290 (0.9)	69 (0.9)	15 (1.2)
Systolic blood pressure (mmHg)	138.5 [127.0, 151.5]	134.5 [123.0, 146.5]	136.5 [125.0, 149.0]
Diastolic blood pressure (mmHg)	78.5 [72.0, 85.5]	78.5 [72.0, 85.0]	78.5 [72.0, 85.5]
Current smoker	980 (3.2)	329 (4.3)	59 (4.6)
Weight (lbs)	164.6 [143.7, 187.7]	164.5 [143.9, 186.6]	177.8 [153.5, 205.2]
Height (in)	66.6 [63.8, 69.3]	66.6 [63.8, 69.0]	67.8 [64.6, 70.7]
BMI (kg/m ²)	25.9 [23.5, 28.8]	26.0 [23.6, 28.9]	26.9 [24.2, 30.2]
Waist circumference (cm)	88.0 [80.0, 97.0]	87.0 [79.0, 95.0]	91.0 [82.0, 99.0]
Hip circumference (cm)	100.0 [95.0, 105.0]	100.0 [96.0, 106.0]	102.0 [97.0, 108.0]
Waist-to-hip ratio	0.88 [0.81, 0.95]	0.87 [0.80, 0.92]	0.89 [0.82, 0.94]
VAT (L)	3.3 [1.9, 5.1]	3.3 [1.9, 4.9]	4.1 [2.4, 6.1]
ASAT (L)	6.3 [4.7, 8.4]	6.4 [4.8, 8.6]	6.6 [4.8, 9.0]
GFAT (L)	9.8 [8.1, 12.0]	9.9 [8.2, 12.0]	10.1 [8.2, 12.4]
Type 2 Diabetes	1488 (4.8)	344 (4.4)	69 (5.4)
Coronary artery disease	1568 (5.1)	328 (4.2)	60 (4.7)

Manuscript Change(s): In response, we have included a new Supplementary Data table stratified by truth label subgroup status, as copied above.

2. Table 1: for variables that are not normally distributed the authors should provide the medians and the interquartile range of the data.

Author Response: We agree that this issue warrants further investigation. We visually inspected distributions of all twelve continuous variables reported in Table 1, copied below:

VAT, ASAT, and GFAT demonstrate a modest right skew, which is reflected in a significant Shapiro-Wilk test (for a random subsample of 5,000 participants: VAT p-value = 5.0×10^{-41} , ASAT p-value = 2.9×10^{-44} , GFAT p-value = 1.3×10^{-37}). To keep the nomenclature consistent throughout the manuscript, we have replaced means and standard deviations with medians and interquartile ranges for all twelve continuous variables.

Manuscript Change(s): In response, we have updated Table 1 to include median with interquartile ranges for continuous variables:

TABLE 1 Baseline characteristics of UK Biobank participants at the time of MRI imaging

	Male (N = 19,435)	Female (N = 20,597)
Age (years)	66.0 [59.3, 71.3]	64.1 [58.0, 69.7]
Self-reported ethnicity		
White	18,773 (96.6)	19,936 (96.8)
Black	137 (0.7)	192 (0.9)
East Asian	112 (0.6)	137 (0.7)
South Asian	238 (1.2)	133 (0.6)
Other	175 (0.9)	199 (1.0)
Systolic blood pressure(mmHg)	140.5 [130.0, 152.5]	134.0 [122.0, 147.5]

Diastolic blood pressure(mmHg)	80.5 [74.0, 87.0]	76.5 [70.0, 83.5]
Current smoker	785 (4.1)	583 (2.9)
Weight (lbs)	181.3 [164.8, 201.3]	147.4 [132.2, 166.3]
Height (in)	69.3 [67.8, 71.1]	64.2 [62.6, 65.8]
Body-mass index (kg/m ²)	26.6 [24.4, 29.1]	25.2 [22.8, 28.5]
Waist circumference (cm)	94.0 [87.0, 101.0]	81.0 [74.0, 90.0]
Hip circumference (cm)	100.0 [96.0, 105.0]	100.0 [94.0, 106.0]
Waist-to-hip ratio	0.93 [0.89, 0.98]	0.81 [0.77, 0.87]
Fat Depot Volumes		
Visceral adipose tissue (L)	4.8 [3.2, 6.4]	2.3 [1.5, 3.5]
Abdominal subcutaneous adipose tissue (L)	5.4 [4.2, 7.0]	7.4 [5.6, 9.7]
Gluteofemoral adipose tissue (L)	8.9 [7.5, 10.7]	10.8 [9.0, 13.1]
Cardiometabolic diseases		
Type 2 diabetes	1,264 (6.5%)	637 (3.1%)
Coronary artery disease	1542 (7.9%)	414 (2.0%)

3. Because about 96.7% of the subjects studied were white, and body fat distribution and the relationships of body fat distribution with cardiometabolic diseases largely differ among different ethnic groups, the authors need to address the limitation of the study that relationships identified in the UK Biobank may not easily apply to other ethnic groups.

Author Response: We agree that the important issue of portability of these results across multiple self-reported ethnicity groups warrants further discussion. Subgroup analyses suggest that the deep learning approach developed in this manuscript performed well across all self-reported ethnicity groups in limited sample size:

Self-reported ethnicity group	Sample size (N) in holdout	Performance; R ² (mean absolute error)
White	VAT: 1,741 ASAT: 1,767 GFAT: 1,499	VAT: 0.992 (0.15) ASAT: 0.991 (0.22) GFAT: 0.978 (0.32)

Black	VAT: 12 ASAT: 11 GFAT: 13	VAT: 0.954 (0.22) ASAT: 0.970 (0.36) GFAT: 0.975 (0.60)
East Asian	VAT: 17 ASAT: 8 GFAT: 8	VAT: 0.991 (0.18) ASAT: 0.998 (0.14) GFAT: 0.995 (0.29)
South Asian	VAT: 21 ASAT: 11 GFAT: 15	VAT: 0.993 (0.11) ASAT: 0.974 (0.25) GFAT: 0.992 (0.34)
Other	VAT: 13 ASAT: 18 GFAT: 16	VAT: 0.996 (0.19) ASAT: 0.998 (0.21) GFAT: 0.990 (0.27)

Unfortunately, we were underpowered to make strong conclusions about disease associations with BMI-adjusted fat depot measures in non-White self-reported ethnicity groups, particularly with coronary artery disease. In general, point estimates of associations with type 2 diabetes were directionally consistent, but point estimates with coronary artery disease were more variable.

Disease	Self-reported ethnicity group	No. Affected/Total No. (%)	VATadjBMI OR (95% CI)	ASATadjBMI OR (95% CI)	GFATadjBMI OR (95% CI)
Prevalent Type 2 Diabetes	White	1694/37411 (4.5%)	1.54 (1.48-1.61)	1.07 (1.02-1.13)	0.75 (0.71-0.79)
	Black	40/311 (12.9%)	1.03 (0.70-1.52)	1.26 (0.89-1.78)	0.62 (0.43-0.90)
	East Asian	23/239 (9.6%)	1.28 (0.79-2.08)	0.88 (0.50-1.56)	1.20 (0.74-1.93)
	South Asian	62/356 (17.4%)	2.06 (1.43-2.97)	1.26 (0.90-1.75)	0.57 (0.37-0.87)
	Other	19/363 (5.2%)	1.30 (0.87-1.94)	0.89 (0.57-1.39)	0.80 (0.50-1.28)
Prevalent Coronary	White	1816/37411 (4.9%)	1.17 (1.11-1.22)	0.99 (0.93-1.05)	0.89 (0.84-0.94)

Artery Disease	Black	8/311 (2.6%)	1.18 (0.57-2.45)	0.81 (0.36-1.79)	1.23 (0.63-2.40)
	East Asian	13/239 (5.4%)	2.42 (0.93-6.27)	0.94 (0.26-3.36)	1.42 (0.42-4.76)
	South Asian	34/356 (9.6%)	0.97 (0.63-1.51)	0.96 (0.63-1.47)	0.98 (0.60-1.59)
	Other	14/363 (3.9%)	1.03 (0.59-1.80)	1.27 (0.68-2.37)	0.77 (0.41-1.46)

We carried out fixed effects meta-analysis across ethnicity subgroups and did not observe evidence of heterogeneity:

	VATadjBMI P-value (ethnicity subgroup heterogeneity)	ASATadjBMI P-value (ethnicity subgroup heterogeneity)	GFATadjBMI P-value (ethnicity subgroup heterogeneity)
Prevalent Type 2 Diabetes	P = 0.83	P = 0.96	P = 0.83
Prevalent Coronary Artery Disease	P = 0.86	P = 0.99	P = 0.95

Manuscript Change(s): In response, we report deep learning model performance and disease associations in self-reported ethnicity subgroups as part of Supplementary Data 4, 10-11. We reference these in the Results sections copied below:

*“Convolutional neural networks – trained on 80% of the participants with fat depots previously quantified – demonstrated near-perfect estimation of each fat depot volume in the 20% of held out individuals ($r^2 = 0.991, 0.991, \text{ and } 0.978$ for VAT, ASAT, and GFAT, respectively) (Supplementary Data 3). **Similar predictive accuracy was noted across age, sex, BMI, and self-reported ethnicity subgroups, although sample size was limited in the latter subgroups (Supplementary Data 4).** These convolutional neural network models were subsequently applied to the remainder of the 40,032 participants to compute fat depot volumes.”*

*“Effect estimates were largely consistent in subgroups binned by age or sex, with somewhat more pronounced magnitude of association in participants with BMI less than 25 (Supplementary Figure 8-9, Supplementary Data 10-12). **Within the limits of statistical power owing to small***

numbers of Black, East Asian, and South Asian participants, we did not detect significant heterogeneity for these associations (p-value for heterogeneity range = 0.83 - 0.96; Supplementary Data 10-11)."

We have also modified the Discussion to discuss this limitation and look ahead to future work:

"Fourth, although our data suggests similar performance of our deep learning models across self-reported ethnicity subgroups, we were underpowered to study disease associations in non-White subgroups. Additional validation across ancestrally and geographically diverse populations would be of considerable value, especially given prior evidence of significant variability in fat distribution indices across ethnicity groups.^{5,6} An important example relates to the South Asian population, where abnormal fat distribution has been postulated as a key driver of the markedly increased rates of cardiovascular disease and diabetes observed, often in the context of a relatively normal BMI.^{7,8}"

4. Sex- and BMI-group specific density plots for VAT, ASAT and GFAT are shown in the figure 2A and prevalence of type 2 diabetes and coronary artery disease, according to quintiles of body-mass index, are shown in the figure 4. Because the East Asians and South Asians studied have different BMI cut-off values for overweight and obesity, than the presented cut-off values, the authors should take this into account in their analyses.

Author Response: We agree that this point warrants further discussion. The reviewer may be referring to "obesity-equivalent" BMI cutoffs that have been proposed in non-European ethnicity groups, including prior work done in the UK Biobank where alternate cutoffs were suggested for BMI = 30 kg/m² equivalents in South Asian, Black, and Chinese participants on the basis of equivalent diabetes risk⁹:

Ethnicity Group	Men (BMI 30 kg/m² equivalent)	Women (BMI 30 kg/m² equivalent)
South Asian	22	22
Black	26	26
Chinese	26	24

We compared the distribution of VAT, ASAT, and GFAT in Black, East Asian (using Chinese cutoffs), and South Asian subgroups first using a BMI = 30 cutoff for obesity, and then using the sex and ethnicity group specific from the above paper. In general, our findings reflect the correlation of BMI with each of the fat depot volumes – using a more conservative cutoff for obesity led to distributions of VAT, ASAT, and GFAT that were centered at lower volumes.

Regarding Figure 4, we do not separately generate predicted prevalence plots by self-reported ethnicity group because estimates of disease associations with BMI-adjusted fat depots in non-White participants had very wide confidence intervals as noted below:

Disease	Self-reported ethnicity group	No. Affected/Total No. (%)	VATadjBMI OR (95% CI)	ASATadjBMI OR (95% CI)	GFATadjBMI OR (95% CI)
Prevalent Type 2 Diabetes	White	1694/37411 (4.5%)	1.54 (1.48-1.61)	1.07 (1.02-1.13)	0.75 (0.71-0.79)
	Black	40/311 (12.9%)	1.03 (0.70-1.52)	1.26 (0.89-1.78)	0.62 (0.43-0.90)
	East Asian	23/239 (9.6%)	1.28 (0.79-2.08)	0.88 (0.50-1.56)	1.20 (0.74-1.93)
	South Asian	62/356 (17.4%)	2.06 (1.43-2.97)	1.26 (0.90-1.75)	0.57 (0.37-0.87)
	Other	19/363 (5.2%)	1.30 (0.87-1.94)	0.89 (0.57-1.39)	0.80 (0.50-1.28)
Prevalent	White	1816/37411	1.17	0.99	0.89

Coronary Artery Disease		(4.9%)	(1.11-1.22)	(0.93-1.05)	(0.84-0.94)
	Black	8/311 (2.6%)	1.18 (0.57-2.45)	0.81 (0.36-1.79)	1.23 (0.63-2.40)
	East Asian	13/239 (5.4%)	2.42 (0.93-6.27)	0.94 (0.26-3.36)	1.42 (0.42-4.76)
	South Asian	34/356 (9.6%)	0.97 (0.63-1.51)	0.96 (0.63-1.47)	0.98 (0.60-1.59)
	Other	14/363 (3.9%)	1.03 (0.59-1.80)	1.27 (0.68-2.37)	0.77 (0.41-1.46)

We carried out fixed effects meta-analysis across ethnicity subgroups and did not observe evidence of heterogeneity:

	VATadjBMI P-value (ethnicity subgroup heterogeneity)	ASATadjBMI P-value (ethnicity subgroup heterogeneity)	GFATadjBMI P-value (ethnicity subgroup heterogeneity)
Prevalent Type 2 Diabetes	P = 0.83	P = 0.96	P = 0.83
Prevalent Coronary Artery Disease	P = 0.86	P = 0.99	P = 0.95

Manuscript Change(s): In response, we have modified the Discussion to include a section discussing the limited evaluation of these results in non-White self-reported ethnicity subgroups:

“Fourth, although our data suggests similar performance of our deep learning models across self-reported ethnicity subgroups, we were underpowered to study disease associations in non-White subgroups. Additional validation across ancestrally and geographically diverse populations would be of considerable value, especially given prior evidence of significant variability in fat distribution indices across ethnicity groups.^{5,6} An important example relates to the South Asian population, where abnormal fat distribution has been postulated as a key driver of the markedly increased rates of cardiovascular disease and diabetes observed, often in the context of a relatively normal BMI.^{7,8}”

5. ASATadjBMI and GFATadjBMI correlated quite strongly with each other in men, and to a much lesser extent in women. The authors should address this point in the discussion and hypothesize about possible mechanisms explaining this difference.

Author Response: We agree that sex differences in the correlation pattern of specific fat depots is of interest. This is an understudied area and to our knowledge no prior study includes abdominal subcutaneous and gluteofemoral adipose tissue volumes quantified at a scale that would enable comparison to the observation in this study. A prior study examining android and gynoid fat percentage (related quantities to ASAT and GFAT, respectively) in NHANES suggested a similar observation in Table 2 (copied from Table 2, with **males in gray**), although further work is needed.¹⁰

	Android fat	Gynoid fat
Android fat (%)	–	0.81*
Gynoid fat (%)	0.70*	–

Manuscript Change(s): In response, we have included this point in the Discussion:

*“Whether individuals in the extreme tails of low GFATadjBMI and ASATadjBMI or high VATadjBMI might be enriched for genetic perturbations in lipodystrophy genes or the inherited component to these metrics is largely ‘polygenic’ – due to the aggregate effects of many common DNA variants, each of modest effect size – warrants further study.^{11–13} **Sex differences will also be important to consider in future studies on local adiposity – for example, here we demonstrate that ASATadjBMI and GFATadjBMI are more correlated in male participants than in female participants, which may point to sex-dependent fat depot specificity.**”*

6. The authors should also discuss why higher VATadjBMI and lower GFATadjBMI most strongly associated with type 2 diabetes and coronary artery disease, in people with a BMI <25, compared to people with overweight or obesity. These relationships may predominantly be driven by a lipodystrophy-like phenotype in people with very low total fat mass.

Author Response: We agree that this is an interesting observation that warrants further discussion. We conducted additional analyses to examine this observation with continuous variables in models including interaction terms between BMI and the BMI-adjusted fat depots and found that this trend is most significant for associations with prevalent type 2 diabetes, but directionally consistent across all analyses.

Supplementary Data 12 Interaction of BMI with BMI-adjusted fat depots

Logistic regression models were used for prevalent diseases and Cox proportional hazards models were used for incident disease. All models included the three BMI-adjusted fat depots, BMI, and interaction terms between BMI and each of the adjusted fat depots. BMI-adjusted fat depots and BMI are in units of sex-specific standard deviations. P-values are reported corresponding to the interaction terms. age, sex, the other two adjusted fat depots, BMI, and MRI imaging center. All models were adjusted for age, sex, and MRI imaging center.

		Beta (Main)	Beta (BMI Interaction)	P-value (BMI interaction)
Prevalent Type 2 Diabetes	VATadjBMI	0.49	-0.09	1.90E-09
	ASATadjBMI	0.11	-0.04	0.04
	GFATadjBMI	-0.39	0.07	7.00E-05
Prevalent Coronary Artery Disease	VATadjBMI	0.15	-0.006	0.77
	ASATadjBMI	0.01	-0.02	0.37
	GFATadjBMI	-0.13	0.008	0.71
Incident Type 2 Diabetes	VATadjBMI	0.49	-0.09	0.008
	ASATadjBMI	-0.04	-0.02	0.69
	GFATadjBMI	-0.21	0.02	0.6
Incident Coronary Artery Disease	VATadjBMI	0.16	-0.01	0.67
	ASATadjBMI	0.05	-0.02	0.56
	GFATadjBMI	-0.13	0.06	0.12

Manuscript Change(s): In response we have added a new supplementary data table as copied above and made reference to this in the Results:

*“In a mutually adjusted logistic regression model including covariates of age, sex, BMI, and MRI assessment center, we observe that VATadjBMI was associated with increased prevalence of type 2 diabetes – OR/SD 1.49; 95% CI 1.43-1.55). By contrast, a largely neutral effect estimate was noted for ASATadjBMI (OR/SD 1.08; 95%CI 1.03-1.14) and GFATadjBMI volumes were associated with decreased risk (OR/SD 0.75; 95% CI: 0.71-0.79) (Figure 3). Effect estimates were largely consistent in subgroups binned by age or sex, with **somewhat more pronounced magnitude of association in participants with BMI less than 25 (Supplementary Figure 8-9, Supplementary Data 10-12).”***

We have also made specific mention of this observation along with a reference to its a similar prior observation in the Discussion:

“Third, changes in measures of local adiposity – independent of weight and body-mass index – may serve as reliable proxies of cardiometabolic benefits of a given intervention, and warrant consideration as additional endpoints for future clinical trials. Most studies to date of obesity interventions have focused on reduction in overall weight or BMI as the primary outcome, consistent with FDA regulatory guidance.¹⁴ However, at least two classes of drugs appear to have

a selective VAT reduction effect in clinical trials: thiazolidinediones and a synthetic form of growth hormone releasing hormone.^{15,16} Whether these therapies might be repurposed from their original indications – type 2 diabetes and HIV-associated lipodystrophy – or new agents might prove useful in a subset of individuals with VAT-driven increases in cardiometabolic risk warrants further study. In such studies, “adjBMI” or similar measures of local adiposity may prove useful for quantifying BMI-independent changes in fat distribution. Considering measures of local adiposity may be particularly important for individuals with normal or low BMI – in this study we observed a trend of amplified associations with type 2 diabetes in participants with BMI less than 25 kg/m², consistent with a prior study examining the association of waist circumference and waist-hip ratio with mortality.¹⁷”

7. In the line 158, shouldn't it read "... in whom probability was 5.3% (95%CI 4.3-6.4).”?

Author Response: We agree that this is a typo. In the revised manuscript, we updated the comparison to compare normal BMI male participants with overweight BMI male participants and confirmed that the correct predicted prevalences were quoted.

Manuscript Change(s): In response, we have updated this section of the Results:

*“As a representative example, males with normal BMI but VATadjBMI in the highest quintile had a probability of type 2 diabetes of 6.6% (95%CI 5.5-7.9), higher than **overweight males with VATadjBMI in the lowest quintile, in whom probability was 2.7% (95%CI 2.2-3.4).**”*

8. Typo in line 125.

Author Response: We agree that this is a typo.

Manuscript Change(s): In response, we have corrected the typo:

*“**Significant correlation between BMI** and all three fat depots was noted – Pearson r ranging from 0.77 to 0.91 – but considerable variation was observed within any clinical BMI category (Figure 2A-B).”*

9. Line 211: the paper referenced as number 45 does not support the statement that "... consistent with FDA regulatory guidance." Also, the papers referenced as 46 and 47 don't support the statement about pharmacological treatment of VAT.

Author Response: We agree that the incorrect citations were linked here. We have corrected this and double-checked the other citations prior to resubmission.

Manuscript Change(s): In response, we have updated the relevant portion of the Discussion:

“Third, changes in measures of local adiposity – independent of weight and body-mass index – may serve as reliable proxies of cardiometabolic benefits of a given intervention, and warrant consideration as additional endpoints for future clinical trials. Most studies to date of obesity interventions have focused on reduction in overall weight or BMI as the primary outcome, consistent with FDA regulatory guidance.¹⁴ However, at least two classes of drugs appear to have a selective VAT reduction effect in clinical trials: thiazolidinediones and a synthetic form of growth hormone releasing hormone.^{15,16} Whether these therapies might be repurposed from their original indications – type 2 diabetes and HIV-associated lipodystrophy – or new agents might prove useful in a subset of individuals with VAT-driven increases in cardiometabolic risk warrants further study.”

Response to Reviewer #2:

Association of machine learning-derived measures of body fat distribution with cardiometabolic diseases in >40,000 individuals

We applaud the authors for developing machine learning approaches to uncover digital readouts of body fat and correlating them with cardio metabolic diseases. Many appreciate the relationship between different measures of adiposity capture different types of metabolic risk, and this approach is one way to dissect this biology. The manuscript can be much improved:

Author Response: We appreciate and agree with the summary above.

1.) Despite the strong correlation with different adiposity measures (Figure 1/2), is the algorithm detecting new biology? UK Biobank contains genetic data, as the authors know well, that could be correlated against their new predictor; the genetic architecture can also be compared between the new predictors and the existing fat measures.

Author Response: We agree that studying the genetic architecture of these and related traits and their genetic correlation with other anthropometric traits is of interest. We carried out these analyses and found that they were too expansive in scope to include in one paper – genetic analyses are present in a follow-up study published in this journal: <https://www.nature.com/articles/s41467-022-30931-2>. Many of our findings in that follow-up study support the observational disease associations found in the present study.

Manuscript Change(s): In response, we have highlighted a section in the Discussion emphasizing the importance of follow-up genetic studies of these and related traits and add a reference to the above study:

“Whether individuals in the extreme tails of low GFATadjBMI and ASATadjBMI or high VATadjBMI might be enriched for genetic perturbations in lipodystrophy genes or the inherited component to these metrics is largely ‘polygenic’ – due to the aggregate effects of many common DNA variants, each of modest effect size – warrants further study.”¹¹⁻¹³”

2.) Related to 1, measures such as diet and physical activity are also available - are these modifiable factors correlated with the new predictors? Are these associations heterogeneous?

Author Response: We agree that studying the association between self-reported diet and physical activity with the newly derived metrics of local adiposity would add to the manuscript. We have added a new section to the results section (copied below) highlighting these associations. Notably, participants with ideal diet tend to have lower VATadjBMI with smaller changes to ASATadjBMI and GFATadjBMI,

while participants with ideal physical activity tend to have relatively symmetrically lower VATadjBMI, ASATadjBMI, and GFATadjBMI. This may be an interesting area for further research.

Supplementary Data 17 Association of healthy diet and physical activity with fat depots

Beta coefficients are in units of sex-specific standard deviations of the fat measurement unless otherwise noted.

Linear regressions included categorical variables for diet (poor/ideal) and physical activity (poor/intermediate/ideal) with poor diet and poor physical activity set as the reference.

All models were adjusted for age, sex, smoking status, and MRI imaging center.

Units: sex-specific SDs	Lifestyle Factor	Category	Beta (95% CI)	P-value
VATadjBMI	Diet	Poor	Ref	Ref
		Ideal	-0.15 (-0.18, -0.13)	6.80E-39
	Physical Activity	Poor	Ref	Ref
		Intermediate	-0.13 (-0.17, -0.09)	5.30E-11
		Ideal	-0.30 (-0.34, -0.26)	1.10E-52
	ASATadjBMI	Diet	Poor	Ref
Ideal			-0.04 (-0.06, -0.02)	1.00E-03
Physical Activity		Poor	Ref	Ref
		Intermediate	-0.07 (-0.11, -0.03)	1.20E-03
		Ideal	-0.27 (-0.31, -0.23)	4.80E-42
GFATadjBMI		Diet	Poor	Ref
	Ideal		-0.03 (-0.05, 0.00)	3.00E-02
	Physical Activity	Poor	Ref	Ref
		Intermediate	-0.08 (-0.12, -0.04)	2.00E-04
		Ideal	-0.27 (-0.31, -0.23)	4.10E-41

Manuscript Change(s): In response, we have added a new section to the Results section with corresponding to Supplementary Data 16-17:

“Of the 40,032 studied participants, 39,530 had self-reported data regarding diet and physical activity available at the time of imaging (Supplementary Data 16). Participants were categorized as following either an ideal or poor diet and either ideal, intermediate, or poor physical activity on the basis of previously defined criteria.¹⁸ We studied associations between diet and physical activity categories with each BMI-adjusted fat depot in linear regressions adjusted for age, sex, smoking status, and MRI assessment center. Ideal diet was associated with reduced VATadjBMI (beta = -0.15 SDs, 95% CI -0.18 - -0.13, P = 6.8 x 10⁻³⁹), with weaker associations noted with ASATadjBMI (beta = -0.04 SDs, 95% CI: -0.06 - -0.02, P = 0.001) and GFATadjBMI (beta = -0.03 SDs, 95% CI: -0.05 - 0.00, P = 0.03) (Supplementary Data 17). Intermediate versus poor physical activity revealed a more symmetric pattern with reduced VATadjBMI (beta = -0.13 SDs, 95% CI: -0.17 - -0.09, P = 5.3 x 10⁻¹¹), ASATadjBMI (beta = -0.07 SDs, 95% CI: -0.11 - -0.03), and GFATadjBMI (beta = -0.08 SDs, 95% CI: -0.12 - -0.04). Ideal versus poor physical activity showed a similar pattern with an amplified effect. Similar patterns were noted in models examining associations with unadjusted VAT, ASAT, and GFAT.”

3.) Little information is provided about the deep learning approach; the parameters used and the methods for cross validation; please include. While “black boxes”, methods such as activation maps can describe what the algorithm is focusing on. This reviewer encourages the use of existing libraries to better understand the predictions and/or the non-concordant data.

Author Response: We agree that additional details about the deep learning approach are warranted. We have moved some of the details pertaining to the deep learning approach and cross-validation scheme from the Supplement to the main Methods (see below in Manuscript Change).

We also agree that one limitation of our approach compared to segmentation is model interpretability; to address this, we have included new saliency mapping analyses on several participants. We applied Grad-CAM to (1) participants with particularly high absolute error in each of the VAT, ASAT, and GFAT held out datasets and (2) participants who were present in all three held out datasets to better understand model predictions. In all cases, we observed anatomically consistent patterns, as shown in the example below:

Supplementary Figure 2 Grad-CAM in female participants with high absolute error

Results from saliency mapping using Grad-CAM are shown for nine female participants among the held out datasets at the 75th, 95th, or 99th percentiles of absolute error of prediction for either VAT, ASAT, or GFAT.

Manuscript Change(s): In response, we include additional details about the deep learning approach and cross-validation scheme in the Methods:

“Individuals with previously quantified fat depot volumes were randomly split into 80% for training and a 20% holdout sample for testing. For each of VAT, ASAT, and GFAT, a CNN was trained on a pair of fat phase and water phase MRI images to predict each fat depot volume, where each image was composed of (a) a coronal two-dimensional projection and (b) a sagittal two-dimensional projection of the body MRI. Each CNN was developed with the DenseNet-121 architecture pre-trained on ImageNet as the base model.^{19,20} The last dense block output was flattened using a global average pooling layer and then fed into three fully connected layers of size 64, 256, and 1, with the last layer having no activation function (linear mapping). All other activation functions use the ReLU non-linearity. All models were trained using the Adam optimizer with a learning rate set to a cosine decay policy decaying from 0.001 to 0 over 100 epochs, a shrinkage loss function using the hyperparameters $a = 10.0$ and $c = 0.2$, and a batch size of 32.^{21,22} For all training data, the following augmentations (random permutations of the training images) were applied: random shifts in the XY-plane by up to ± 16 pixels, rotations by up to ± 5 degrees around its center axis, and the coronal view horizontally flipped with a probability of 50%. Each view (coronal and sagittal) were separately pre-normalized by its z-score (0 mean, standard deviation of 1), followed by joint normalization following concatenation side-by-side.

A five-fold cross-validation scheme was used within the 80% training data set. Performance was determined in a 20% holdout sample that was unseen to the model prior to evaluation. The five folds were used to determine mean and standard deviation of performance metrics, then a single fold was randomly selected to take forward for predicting fat depot volumes in the remaining participants with raw MRI imaging data but without labels. More comprehensive descriptions of the deep learning modeling and quality control are provided in the Supplementary Methods.”

We have also added a new paragraph to the Results discussing Grad-CAM saliency mapping:

“We applied Grad-CAM to better understand regions of a given MRI projection contributing to predictions of VAT, ASAT, and GFAT volumes.²³ Separately in males and females, we selected participants from each held out dataset at the 75th, 95th, and 99th percentiles of absolute error and applied Grad-CAM. We also selected three participants who were present in all three held out datasets to compare Grad-CAM results for VAT, ASAT, and GFAT. In all cases, Grad-CAM revealed prioritized regions of the MRI projection that were anatomically consistent with the known distribution of VAT, ASAT, and GFAT, even in cases with higher absolute error (Supplementary Figures 1-4).”

4.) The authors leave a lot of room to describe clinical utility of these new readouts: what is meant by “additional endpoints for obesity interventions...” Can the authors describe a study to examine how

interventions might reflect changes in these image readouts that would be different than that of measuring the phenotypes they trained on?

Author Response: We agree that additional clarification surrounding the potential clinical utility of these metrics is warranted. The “adjBMI” paradigm presented here is one way to make a continuous variable to quantify the notion of “fat depot changes out of proportion to BMI”. In the simplest example of this, a drug may leave BMI unchanged following administration but lead to selective reduction in VAT – a previously published trial of tesamorelin in patients with HIV lipodystrophy is an example of this.¹⁶ In this case, there is a clear BMI-independent change in VAT without the need for any alteration to the phenotypes. In a more complex situation, however, where an intervention is leading to complex changes in both BMI and specific fat depots, we propose that the “adjBMI” paradigm is one simple way to discuss these changes, acknowledging that it has several limitations.

Manuscript Change(s): In response, we have added some clarifying wording to the Discussion:

*“However, at least two classes of drugs appear to have a selective VAT reduction effect in clinical trials: thiazolidinediones and a synthetic form of growth hormone releasing hormone.^{15,16} Whether these therapies might be repurposed from their original indications – type 2 diabetes and HIV-associated lipodystrophy – or new agents might prove useful in a subset of individuals with VAT-driven increases in cardiometabolic risk warrants further study. **In such studies, “adjBMI” or similar measures of local adiposity may prove useful for quantifying BMI-independent changes in fat distribution. Considering measures of local adiposity may be particularly important for individuals with normal or low BMI – in this study we observed a trend of amplified associations with type 2 diabetes in participants with BMI less than 25 kg/m², consistent with a prior study examining the association of waist circumference and waist-hip ratio with mortality.¹⁷”***

5.) The authors mention limited age windows of surveillance; however, others have shown the ability for modalities such as MRIs to predict age in the UK Biobank (Le Goallec 2021, Nature Comm; Jonsson ,2019) and may be residually confounded. Have the authors considered a predictor that stratifies by age to get at question posed in 1.)?

Author Response: We agree that predictive performance across age subgroups of interest. To address this, we stratified the holdout data sets by age for each of the VAT, ASAT, and GFAT models and reported performance within subgroups. Performance remained high across all subgroups:

Supplementary Data 4 Convolutional neural network performance in subgroups

Predictive performance is displayed for VAT, ASAT, and GFAT for age, sex, BMI, and self-reported ethnicity subgroups.

In this table, we report performance for the arbitrarily selected fold used for prediction in unlabeled participants to align with Figure 1 (see Methods).

Abbreviations: MAE, mean absolute error.

		VAT			ASAT			GFAT		
		N	R2	MAE	N	R2	MAE	N	R2	MAE
Age	All	1804	0.991	0.147	1815	0.991	0.217	1551	0.978	0.318
	Under 60	601	0.993	0.135	638	0.993	0.214	568	0.98	0.317
	60 to 70	874	0.992	0.148	840	0.991	0.214	725	0.975	0.323
	Over 70	329	0.987	0.167	337	0.986	0.23	258	0.982	0.309

Manuscript Change(s): In response, we have updated the Results to make note of predictive performance across age subgroups:

“Convolutional neural networks – trained on 80% of the participants with fat depots previously quantified – demonstrated near-perfect estimation of each fat depot volume in the 20% of held out individuals ($r^2 = 0.991, 0.991, \text{ and } 0.978$ for VAT, ASAT, and GFAT, respectively) (Supplementary Data 3). Similar predictive accuracy was noted across age, sex, BMI, and self-reported ethnicity subgroups, although sample size was limited in the latter subgroups (Supplementary Data 4).”

We have also made note of this reviewer’s observation that abdominal imaging may have utility for learning hidden variables such as age in the Discussion:

“Although population-based assessment of fat distribution using MRI is unlikely to be practical, these results lay the scientific foundation for efforts to quantify such measures using other data – such as DEXA images or abdominal CT scans already embedded in the electronic medical record for some patients – or even static images of body silhouette, as might conceivably be obtained with a smartphone camera.^{24,25} Abdominal imaging may also be useful for learning hidden variables of biological significance, such as age.²⁶”

Response to Reviewer #3:

In their manuscript entitled "Association of machine learning-derived measures of body fat distribution with cardiometabolic diseases in >40,000 individuals" the authors describe the results of the analysis of adipose tissue distribution from MRI data in participants from the UK Biobank study.

Overall, the topic of this work - the analysis of large-scale epidemiological imaging data - is highly relevant and important for the scientific community as well as potentially for medical practice.

In this work, the authors provide mainly provide the following two core contributions:

- (i) simplification and optimization of automated quantification of adipose tissue compartments
- (ii) correlation of adipose tissue distribution with a limited number of clinical endpoints, specifically the presence of type 2 diabetes.

Overall the manuscript is clearly structured and well-written.

Author Response: We appreciate and agree with the summary above.

However, there are major and minor limitations that need to be addressed:

Major points:

I would like to encourage the authors to perform more detailed and thorough analyses. The large-scale data the authors were able to access offers a unique opportunity for scientific analyses but also demands a particular level of analytic effort. This includes:

1) The statistical analyses presented in this work remain partly superficial and inconsistent. First, the authors treat different factors of variance differently without clear reason. Thus, BMI is directly adjusted for in adjusted adipose tissue volumes, but potentially equally important factors such as age are only considered as subgroups. It could e.g. very well be that the computed VATadjBMI just reflects changes of VAT with age and is therefore only indirectly associated with type 2 diabetes (with age as a confounder). Other factors, such as racial background, body height and weight (BMI as an aggregate measure loses some information), may similarly act as confounding factors.

Only after correction for these and other potential factors of variation is it possible to hypothesize an independent association between adipose tissue distribution and cardiovascular disease.

Author Response: We agree that careful statistical analysis is crucial for generating hypotheses about the associations between adipose tissue distribution and cardiometabolic diseases. We clarify that the core analyses reported in the manuscript were all adjusted for age, sex, BMI, the other two fat depots (e.g. ASATadjBMI and GFATadjBMI for VATadjBMI), and MRI imaging center.

We agree that other confounding variables are possible and so repeated the primary analyses with models additionally adjusted for weight, height, smoking status, and self-reported ethnicity. Effect sizes were modestly changed, and in general we observed good agreement:

Supplementary Data 13 Association with prevalent cardiometabolic diseases: sensitivity analyses						
Model 1 is the primary model used in the paper including the three BMI-adjusted fat depot volumes, age, sex, BMI, and MRI imaging center.						
Model 2 starts with Model 1 and adds weight, height, self-reported ethnicity, and smoking status.						
Model 3 starts with Model 1 and adds prevalent type 2 diabetes status.						
Type 2 Diabetes						
	VATadjBMI OR (95% CI)	P-value	ASATadjBMI OR (95% CI)	P-value	GFATadjBMI OR (95% CI)	P-value
Model1	1.49 (1.43-1.55)	9.90E-76	1.08 (1.03-1.14)	0.002	0.75 (0.71-0.79)	6.40E-28
Model2	1.53 (1.46-1.60)	3.40E-74	1.07 (1.01-1.12)	0.013	0.75 (0.71-0.80)	1.10E-21
Coronary Artery Disease						
	VATadjBMI OR (95% CI)	P-value	ASATadjBMI OR (95% CI)	P-value	GFATadjBMI OR (95% CI)	P-value
Model1	1.17 (1.11-1.22)	3.00E-11	1.00 (0.94-1.05)	0.921	0.89 (0.84-0.94)	3.50E-05
Model2	1.20 (1.14-1.26)	1.40E-13	1.00 (0.94-1.06)	0.956	0.94 (0.88-1.00)	0.051
Model3	1.14 (1.09-1.19)	3.20E-08	0.99 (0.94-1.05)	0.773	0.91 (0.86-0.96)	5.50E-04

Manuscript Change(s): In response, we have modified the relevant portion of the Results section to (1) clarify the covariates in the primary analyses of the manuscript and (2) include results of a sensitivity analysis including additional covariates:

*“In contrast to analysis of raw tissue volumes – where each depot was associated with increased risk – significant heterogeneity was noted for BMI-adjusted values. **In a mutually adjusted logistic regression model including covariates of age, sex, BMI, and MRI assessment center, we observe that VATadjBMI was associated with increased prevalence of type 2 diabetes – OR/SD 1.49; 95% CI 1.43-1.55). By contrast, a largely neutral effect estimate was noted for ASATadjBMI (OR/SD 1.08; 95%CI 1.03-1.14) and GFATadjBMI volumes were associated with decreased risk (OR/SD 0.75; 95% CI: 0.71-0.79) (Figure 3). Effect estimates were largely consistent in subgroups binned by age or sex, with somewhat more pronounced magnitude of association in participants with BMI less than 25 (Supplementary Figure 8-9, Supplementary Data 10-12). We did not observe heterogeneity in disease association across self-reported ethnicity subgroups, but were underpowered in Black, East Asian, and South Asian subgroups (Supplementary Data 10-11). A similar pattern was observed for coronary artery disease, where associations for VATadjBMI, ASATadjBMI, and GFATadjBMI were 1.17 (95%CI 1.11-1.22), 1.00 (95%CI 0.94-1.05), and 0.89 (95%CI 0.84-0.94), respectively. In a sensitivity analysis, we additionally adjusted for weight, height, smoking status, and self-reported ethnicity, finding similar results (Supplementary Data 13). Adjustment for type 2 diabetes status in the coronary artery disease analysis led to comparable results as well.”***

2) While the simplification of adipose tissue quantification made by the authors (from 3D to 2D projections) may allow for faster analyses, it has the major disadvantage that crucial information that is present in the image volumes is discarded. In the context of this work, the used imaging data in principle allow for quantification of further adipose tissue compartments including, e.g., pericardial adipose tissue, kidney hilum adipose tissue as well as liver and pancreas fat content. The analysis of data from

large-scale population studies such as the UK Biobank should utilize all relevant information that is available in order to provide a complete picture of the data. The additional effort is justified considering the potential impact of these results.

Thus, although technically interesting, the proposed simplified analysis can be seen as an oversimplification.

Author Response: We agree that one key area of future work is to parametrize other areas of fat and lean tissue across the human body. Prior efforts with the goal of quantifying fat in other regions such as the liver, pancreas, and epicardial region, have yielded promising results.²⁷⁻²⁹ In this work, we focused on quantifying three adipose tissue compartment volumes of cardiometabolic significance at scale – 2D projections enabled near-perfect performance for prediction of these quantities. An alternative goal that aims to use a single imaging input to simultaneously predict the “volume quantities” in this paper alongside “density-like quantities” like liver fat % may benefit from an alternate approach to image-simplification.

Manuscript Change(s): In response, we have modified the Discussion to highlight this area for further research:

*“Our study has several limitations. First, this study was a cross-sectional analysis of individuals with a median age of 65 years at time of imaging. Future studies of individuals across the lifespan – especially those that include repeat imaging assessments – are warranted. Second, although we note striking associations of BMI-adjusted fat depots with cardiometabolic disease, these observational data do not definitely prove causation or that modification of fat distribution will lead to therapeutic gain. **Third, while two-dimensional MRI projections are a useful simplification of three-dimensional MRI images for the task of predicting adipose tissue compartment volumes, they are unlikely to be appropriate for predicting “density-like quantities” such as liver fat percentage, where a single axial cross-section performs well.**”²⁹ This highlights the importance of choosing an appropriate simplification for the desired task.”*

Further points:

3) Title: the data source (UK Biobank) should be mentioned in the title

Author Response: We agree that the data source is crucial to communicate early in the manuscript. To limit the length of the title, we highlight the UK Biobank data source early in the Abstract:

*“Methods: **We studied MRI imaging data of 40,032 UK Biobank participants.** Using previously quantified visceral (VAT), abdominal subcutaneous (ASAT), and gluteofemoral (GFAT) adipose tissue volume in up to 9,041 to train convolutional neural networks (CNNs), we quantified these depots in the remainder of the participants.”*

Manuscript Change(s): In addition to mention in the abstract above, we have modified the wording of the first sentence of the Results:

*“Among **40,032 participants of the UK Biobank** with MRI data available, the median age was 65 years, 51% were female, and 97% were white (Table 1). Median BMI was 26.6 kg/m² among males and 25.2 kg/m² among females, and median waist-hip ratio (WHR) was 0.93 among males, and 0.81 among females.”*

4) Methods: How was quality assurance performed on the test set? The proposed adipose tissue quantification method - in contrast to segmentation-based methods - has the drawback of limited interpretability. How were image volumes with wrong automated adipose tissue estimates identified?

Author Response: We agree that one limitation of our approach compared to segmentation is model interpretability; to address this, we have included new saliency mapping analyses on several participants. We applied Grad-CAM to (1) participants with particularly high absolute error in each of the VAT, ASAT, and GFAT held out datasets and (2) participants who were present in all three held out datasets to better understand model predictions. In all cases, we observed anatomically consistent patterns, as shown in the example below:

Supplementary Figure 2 Grad-CAM in female participants with high absolute error

Results from saliency mapping using Grad-CAM are shown for nine female participants among the held out datasets at the 75th, 95th, or 99th percentiles of absolute error of prediction for either VAT, ASAT, or GFAT.

Manuscript Change(s): We have added a new paragraph to the Results discussing Grad-CAM saliency mapping:

“We applied Grad-CAM to better understand regions of a given MRI projection contributing to predictions of VAT, ASAT, and GFAT volumes.²³ Separately in males and females, we selected participants from each held out dataset at the 75th, 95th, and 99th percentiles of absolute error and applied Grad-CAM. We also selected three participants who were present in all three held out datasets to compare Grad-CAM results for VAT, ASAT, and GFAT. In all cases, Grad-CAM revealed prioritized regions of the MRI projection that were anatomically consistent with the known distribution of VAT, ASAT, and GFAT, even in cases with higher absolute error (Supplementary Figures 1-4).”

We have also added a point in the Discussion highlighting the limitation that despite good performance and consistent results from saliency mapping, we did not directly compare our approach to a segmentation model:

*“Our study has several limitations. First, this study was a cross-sectional analysis of individuals with a median age of 65 years at time of imaging. Future studies of individuals across the lifespan – especially those that include repeat imaging assessments – are warranted. Second, although we note striking associations of BMI-adjusted fat depots with cardiometabolic disease, these observational data do not definitely prove causation or that modification of fat distribution will lead to therapeutic gain. Third, while two-dimensional MRI projections are a useful simplification of three-dimensional MRI images for the task of predicting adipose tissue compartment volumes, they are unlikely to be appropriate for predicting “density-like quantities” such as liver fat percentage, where a single axial cross-section performs well.²⁹ This highlights the importance of choosing an appropriate simplification for the desired task. **Fourth, while we were able to achieve good performance using CNN-based regression models and saliency mapping results were anatomically reasonable, we were unable to directly compare our approach to segmentation-based models.**”*

5) Results: Did the authors have access to more detailed temporal information regarding the onset of cardiovascular disease. It would be of high interest to look into the correlation between adipose tissue volumes and disease progression if possible.

Author Response: We agree that the association of adipose tissue distribution with time to disease is of high interest. While limited by sample size, we use Cox proportional hazards models to test the association of BMI-adjusted fat depot volumes with incident type 2 diabetes and coronary artery disease and find good agreement with prevalent disease analyses:

TABLE 2 Association of BMI-adjusted fat depot volumes with incident disease

Disease	No. Events / Total No. at Risk (%)	BMI-adjusted Fat Depot	HR (95% CI)	P-value
Type 2 Diabetes	235/37,891 (0.6)	VATadjBMI	1.45 (1.30 - 1.61)	1.3E-11
		ASATadjBMI	0.96 (0.84 - 1.08)	0.49
		GFATadjBMI	0.84 (0.74 - 0.95)	0.005
Coronary artery disease	607/38,067 (1.6)	VATadjBMI	1.17 (1.08 - 1.26)	8.1E-05
		ASATadjBMI	1.04 (0.95 - 1.14)	0.41
		GFATadjBMI	0.91 (0.83 - 1.00)	0.05

Hazard ratios with 95% CI in parentheses are shown for VATadjBMI, ASATadjBMI, and GFATadjBMI in Cox proportional-hazard models adjusted for age, sex, BMI, the other two fat depots, and MRI imaging center. Median follow-up time for both incident type 2 diabetes and coronary artery disease was 2.8 years from the date of imaging. Note that two participants in the prevalent disease analyses are not included in incident disease analyses because they withdrew consent in the interim period.

Manuscript Change(s): In response, we have moved the results from the incident disease associations from the supplement to the main text as Table 2:

TABLE 2 Association of BMI-adjusted fat depot volumes with incident disease

Disease	No. Events / Total No. at Risk (%)	BMI-adjusted Fat Depot	HR (95% CI)	P-value
Type 2 Diabetes	235/37,891 (0.6)	VATadjBMI	1.45 (1.30 - 1.61)	1.3E-11
		ASATadjBMI	0.96 (0.84 - 1.08)	0.49
		GFATadjBMI	0.84 (0.74 - 0.95)	0.005
Coronary artery disease	607/38,067 (1.6)	VATadjBMI	1.17 (1.08 - 1.26)	8.1E-05
		ASATadjBMI	1.04 (0.95 - 1.14)	0.41
		GFATadjBMI	0.91 (0.83 - 1.00)	0.05

Hazard ratios with 95% CI in parentheses are shown for VATadjBMI, ASATadjBMI, and GFATadjBMI in Cox proportional-hazard models adjusted for age, sex, BMI, the other two fat depots, and MRI imaging center. Median follow-up time for both incident type 2 diabetes and coronary artery disease was 2.8 years from the date of imaging. Note that two participants in the prevalent disease analyses are not included in incident disease analyses because they withdrew consent in the interim period.

6) Figures: Figure 1C is somehow misleading as it suggest segmentation of adipose tissue compartments, which is however not done by the proposed algorithm. It should be clarified that the provided regions were hand-drawn for illustration purposes.

Author Response: We agree that the visual aid around each fat depot may be misleading as the proposed algorithm did not utilize segmented fat depots. We have added a note in the figure legend to clarify this potential misunderstanding.

Manuscript Change(s): In response, we have modified the Figure 1 legend:

*“(A) Two-dimensional projections are created by computing the mean pixel intensity along the coronal (top) and sagittal (bottom) axes. Two images were used as inputs into the convolutional neural network: one consisting of the coronal and sagittal two-dimensional projections in the fat phase, and another consisting of the same projections in the water phase. (B) Convolutional neural networks trained on two-dimensional MRI projections achieved near-perfect prediction of each fat depot volume in the holdout set (Supplementary Table 3). (C) Three female participants with similar BMI (ranging from 29.1 to 29.6 kg/m²) but highly discordant fat depot volumes quantified by convolutional neural networks. Fat depot volume percentiles are computed relative to a subgroup of female participants with overweight BMI (25 ≤ BMI < 30). **Note that outlines for***

each fat depot are drawn as a visual aid for each fat depot and do not reflect segmentation. Abbreviations: VAT, visceral adipose tissue; ASAT, abdominal subcutaneous adipose tissue; GFAT, gluteofemoral adipose tissue.”

7) Methods: It could be interesting to look into gradient-based saliency maps of the adipose tissue segmentation models in order to assess whether estimates are based on the respective anatomic regions or maybe based on completely different information.

Author Response: We agree and have included new saliency mapping analyses on several participants. We applied Grad-CAM to (1) participants with particularly high absolute error in each of the VAT, ASAT, and GFAT held out datasets and (2) participants who were present in all three held out datasets to better understand model predictions. In all cases, we observed anatomically consistent patterns, as shown in the example below:

Supplementary Figure 2 Grad-CAM in female participants with high absolute error

Results from saliency mapping using Grad-CAM are shown for nine female participants among the held out datasets at the 75th, 95th, or 99th percentiles of absolute error of prediction for either VAT, ASAT, or GFAT.

Manuscript Change(s): We have added a new paragraph to the Results discussing Grad-CAM saliency mapping:

“We applied Grad-CAM to better understand regions of a given MRI projection contributing to predictions of VAT, ASAT, and GFAT volumes.²³ Separately in males and females, we selected participants from each held out dataset at the 75th, 95th, and 99th percentiles of absolute error and applied Grad-CAM. We also selected three participants who were present in all three held out datasets to compare Grad-CAM results for VAT, ASAT, and GFAT. In all cases, Grad-CAM revealed prioritized regions of the MRI projection that were anatomically consistent with the known distribution of VAT, ASAT, and GFAT, even in cases with higher absolute error (Supplementary Figures 1-4).”

8) Methods: What justifies linear adjustment of adipose tissue volumes for BMI? Was a linear relationship observed?

Authors Reply: We agree that further discussion around the choice of a linear adjustment with respect to BMI is warranted. Across the majority of the BMI spectrum, we observed good agreement between a linear fit and a spline fit using a B-spline basis with knots at BMI = 25, 30, and 35 kg/m².

Residuals from these two fits showed good agreement:

Supplementary Data 9 Correlation between BMI residuals from a linear fit versus spline fit

Pearson correlations are shown between residuals from a linear fit of each fat depot volume versus BMI, compared to a spline fit. The spline fit used a B-spline basis with knots at BMI = 25, 30, and 35 kg/m².

	Male	Female
VATadjBMI	0.983	0.991
ASATadjBMI	0.983	0.993
GFATadjBMI	0.996	0.999

Manuscript Change(s): In response, we have added a new supplementary figure and supplementary table showing the above data and included a comment in the Results:

“To disentangle the unique impact of each fat depot from overall BMI, we next generated new measurements of VATadjBMI, ASATadjBMI, and GFATadjBMI for each participant by computing sex-specific BMI residuals in 38,680 (97%) of the study population with BMI measurement on the day of MRI imaging available. These residuals reflect the difference in an individual’s fat depot volume as compared with that expected based on BMI. These metrics were fully independent of BMI and largely independent of anthropometric measures and each other (Supplementary Figure 6). Flexibly modeling BMI with a B-spline basis when computing these residuals yielded similar results (Supplementary Figure 7, Supplementary Data 9).”

9) Methods: The observed clinical endpoints, specifically type 2 diabetes may themselves be a confounder for associations with other clinical endpoints (e.g. the association of VAT depots with cardiac disease may be mediated by type two diabetes). These potential effects should be considered in the statistical analysis.

Authors Reply: We agree that type 2 diabetes could be a confounder for the association between adipose tissue distribution and coronary artery disease. To test this, we additionally adjusted prevalent coronary artery disease analyses for type 2 diabetes status and found similar results:

Supplementary Data 13 Association with prevalent cardiometabolic diseases: sensitivity analyses						
Model 1 is the primary model used in the paper including the three BMI-adjusted fat depot volumes, age, sex, BMI, and MRI imaging center.						
Model 2 starts with Model 1 and adds weight, height, self-reported ethnicity, and smoking status.						
Model 3 starts with Model 1 and adds prevalent type 2 diabetes status.						
Type 2 Diabetes						
	VATadjBMI OR (95% CI)	P-value	ASATadjBMI OR (95% CI)	P-value	GFATadjBMI OR (95% CI)	P-value
Model1	1.49 (1.43-1.55)	9.90E-76	1.08 (1.03-1.14)	0.002	0.75 (0.71-0.79)	6.40E-28
Model2	1.53 (1.46-1.60)	3.40E-74	1.07 (1.01-1.12)	0.013	0.75 (0.71-0.80)	1.10E-21
Coronary Artery Disease						
	VATadjBMI OR (95% CI)	P-value	ASATadjBMI OR (95% CI)	P-value	GFATadjBMI OR (95% CI)	P-value
Model1	1.17 (1.11-1.22)	3.00E-11	1.00 (0.94-1.05)	0.921	0.89 (0.84-0.94)	3.50E-05
Model2	1.20 (1.14-1.26)	1.40E-13	1.00 (0.94-1.06)	0.956	0.94 (0.88-1.00)	0.051
Model3	1.14 (1.09-1.19)	3.20E-08	0.99 (0.94-1.05)	0.773	0.91 (0.86-0.96)	5.50E-04

Manuscript Change(s): In response, we have made note of this sensitivity analysis in the Results:

“In contrast to analysis of raw tissue volumes – where each depot was associated with increased risk – significant heterogeneity was noted for BMI-adjusted values. In a mutually adjusted logistic regression model including covariates of age, sex, BMI, and MRI assessment center, we observe that VATadjBMI was associated with increased prevalence of type 2 diabetes – OR/SD 1.49; 95% CI 1.43-1.55). By contrast, a largely neutral effect estimate was noted for ASATadjBMI (OR/SD 1.08; 95%CI 1.03-1.14) and GFATadjBMI volumes were associated with decreased risk (OR/SD 0.75; 95% CI: 0.71-0.79) (Figure 3). Effect estimates were largely consistent in subgroups binned by age or sex, with somewhat more pronounced magnitude of association in participants with BMI less than 25 (Supplementary Figure 8-9, Supplementary Data 10-12). We did not observe heterogeneity in disease association across self-reported ethnicity subgroups, but were underpowered in Black, East Asian, and South Asian subgroups (Supplementary Data 10-11). A

*similar pattern was observed for coronary artery disease, where associations for VATadjBMI, ASATadjBMI, and GFATadjBMI were 1.17 (95%CI 1.11-1.22), 1.00 (95%CI 0.94-1.05), and 0.89 (95%CI 0.84-0.94), respectively. In a sensitivity analysis, we additionally adjusted for weight, height, smoking status, and self-reported ethnicity, finding similar results (Supplementary Data 13). **Adjustment for type 2 diabetes status in the coronary artery disease analysis led to comparable results as well.***

REFERENCES

1. West J, Leinhard OD, Romu T, et al. Feasibility of MR-Based Body Composition Analysis in Large Scale Population Studies. *PLOS ONE* 2016;11(9):e0163332.
2. Borga M, West J, Bell JD, et al. Advanced body composition assessment: from body mass index to body composition profiling. *J Investig Med Off Publ Am Fed Clin Res* 2018;66(5):1–9.
3. Linge J, Borga M, West J, et al. Body Composition Profiling in the UK Biobank Imaging Study. *Obes Silver Spring Md* 2018;26(11):1785–95.
4. Linge J, Witcher B, Borga M, Dahlqvist Leinhard O. Sub-phenotyping Metabolic Disorders Using Body Composition: An Individualized, Nonparametric Approach Utilizing Large Data Sets. *Obes Silver Spring Md* 2019;27(7):1190–9.
5. Kanaley JA, Giannopoulou I, Tillapaugh-Fay G, Nappi JS, Ploutz-Snyder LL. Racial differences in subcutaneous and visceral fat distribution in postmenopausal black and white women. *Metabolism* 2003;52(2):186–91.
6. Shah Ravi V., Murthy Venkatesh L., Abbasi Siddique A., et al. Visceral Adiposity and the Risk of Metabolic Syndrome Across Body Mass Index. *JACC Cardiovasc Imaging* 2014;7(12):1221–35.
7. Raji A, Seely EW, Arky RA, Simonson DC. Body fat distribution and insulin resistance in healthy Asian Indians and Caucasians. *J Clin Endocrinol Metab* 2001;86(11):5366–71.
8. Patel AP, Wang M, Kartoun U, Ng K, Khera AV. Quantifying and Understanding the Higher Risk of Atherosclerotic Cardiovascular Disease Among South Asian Individuals: Results From the UK Biobank Prospective Cohort Study. *Circulation* 2021;144(6):410–22.
9. Ntuk UE, Gill JMR, Mackay DF, Sattar N, Pell JP. Ethnic-Specific Obesity Cutoffs for Diabetes Risk: Cross-sectional Study of 490,288 UK Biobank Participants. *Diabetes Care* 2014;37(9):2500–7.
10. Min K-B, Min J-Y. Android and gynoid fat percentages and serum lipid levels in United States adults. *Clin Endocrinol (Oxf)* 2015;82(3):377–87.
11. Karlsson T, Rask-Andersen M, Pan G, et al. Contribution of genetics to visceral adiposity and its relation to cardiovascular and metabolic disease. *Nat Med* 2019;25(9):1390–5.
12. Lotta LA, Gulati P, Day FR, et al. Integrative genomic analysis implicates limited peripheral adipose storage capacity in the pathogenesis of human insulin resistance. *Nat Genet* 2017;49(1):17–26.
13. Agrawal S, Wang M, Klarqvist MDR, et al. Inherited basis of visceral, abdominal subcutaneous and gluteofemoral fat depots. *Nat Commun* 2022;13(1):3771.
14. Colman Eric. Food and Drug Administration’s Obesity Drug Guidance Document. *Circulation* 2012;125(17):2156–64.
15. Kodama N, Tahara N, Tahara A, et al. Effects of Pioglitazone on Visceral Fat Metabolic Activity in Impaired Glucose Tolerance or Type 2 Diabetes Mellitus. *J Clin Endocrinol Metab* 2013;98(11):4438–45.
16. Stanley TL, Feldpausch MN, Oh J, et al. Effect of tesamorelin on visceral fat and liver fat in HIV-infected patients with abdominal fat accumulation: a randomized clinical trial. *JAMA* 2014;312(4):380–9.
17. Pischon T, Boeing H, Hoffmann K, et al. General and abdominal adiposity and risk of death in Europe. *N Engl J Med* 2008;359(20):2105–20.
18. Said MA, Verweij N, van der Harst P. Associations of Combined Genetic and Lifestyle Risks With Incident Cardiovascular Disease and Diabetes in the UK Biobank Study. *JAMA Cardiol* 2018;3(8):693–702.
19. Huang G, Liu Z, Van Der Maaten L, Weinberger KQ. Densely Connected Convolutional

- Networks. In: 2017 IEEE Conference on Computer Vision and Pattern Recognition (CVPR). 2017. p. 2261–9.
20. Deng J, Dong W, Socher R, Li L, Kai Li, Li Fei-Fei. ImageNet: A large-scale hierarchical image database. In: 2009 IEEE Conference on Computer Vision and Pattern Recognition. 2009. p. 248–55.
 21. Kingma DP, Ba J. Adam: A Method for Stochastic Optimization. ArXiv14126980 Cs [Internet] 2017 [cited 2021 Apr 20]; Available from: <http://arxiv.org/abs/1412.6980>
 22. Lu X, Ma C, Ni B, Yang X, Reid I, Yang M-H. Deep Regression Tracking with Shrinkage Loss [Internet]. 2018 [cited 2021 Apr 20]. p. 353–69. Available from: https://openaccess.thecvf.com/content_ECCV_2018/html/Xiankai_Lu_Deep_Regression_Tracking_ECCV_2018_paper.html
 23. Selvaraju RR, Cogswell M, Das A, Vedantam R, Parikh D, Batra D. Grad-CAM: Visual Explanations from Deep Networks via Gradient-Based Localization. *Int J Comput Vis* 2020;128(2):336–59.
 24. Larson DB, Johnson LW, Schnell BM, Salisbury SR, Forman HP. National Trends in CT Use in the Emergency Department: 1995–2007. *Radiology* 2011;258(1):164–73.
 25. Hu P, Kaashki NN, Dadarlat V, Munteanu A. Learning to Estimate the Body Shape Under Clothing From a Single 3-D Scan. *IEEE Trans Ind Inform* 2021;17(6):3793–802.
 26. Le Goallec A, Diai S, Collin S, Prost J-B, Vincent T, Patel CJ. Using deep learning to predict abdominal age from liver and pancreas magnetic resonance images. *Nat Commun* 2022;13(1):1979.
 27. Liu Y, Bastý N, Witcher B, et al. Genetic architecture of 11 organ traits derived from abdominal MRI using deep learning. *eLife* 2021;10:e65554.
 28. Eisenberg E, McElhinney PA, Commandeur F, et al. Deep Learning–Based Quantification of Epicardial Adipose Tissue Volume and Attenuation Predicts Major Adverse Cardiovascular Events in Asymptomatic Subjects. *Circ Cardiovasc Imaging* 2020;13(2):e009829.
 29. Haas ME, Pirruccello JP, Friedman SN, et al. Machine learning enables new insights into genetic contributions to liver fat accumulation. *Cell Genomics* 2021;1(3):100066.

Reviewers' Comments:

Reviewer #1:

Remarks to the Author:

The authors have very well addressed the critical points and provide important novel information in this field of research.

Norbert Stefan

Reviewer #2:

Remarks to the Author:

No further comments; authors have done great work in this reviewers opinion.

Reviewer #3:

Remarks to the Author:

The authors have provided satisfactory clarifications and additional results as part of their manuscript revision. I do not have further comments.

Response to Referees' Comments for *NCOMMS-22-30756A*
"Association of machine learning-derived measures of body fat distribution with cardiometabolic diseases in >40,000 individuals"

Response to Reviewer #1:

The authors have very well addressed the critical points and provide important novel information in this field of research.

Norbert Stefan

Author Response: Thank you for your helpful comments.

Response to Reviewer #2:

No further comments; authors have done great work in this reviewers opinion.

Author Response: Thank you for your helpful comments.

Response to Reviewer #3:

The authors have provided satisfactory clarifications and additional results as part of their manuscript revision. I do not have further comments.

Author Response: Thank you for your helpful comments.